# Valorization of various lignocellulosic wastes to *Ganoderma lucidum* (Curtis) P. Karst (Reishi Mushroom) cultivation and their FT-IR assessments

Caglar Akcay[1,2], Recai Arslan [ID][2]*, Faik Ceylan [ID][2]

**1** Düzce University, Forestry Vocational School, Department of Forestry, Düzce, Türkiye, **2** Düzce University, Recycling of Agricultural Wastes to Industry Application and Research Center (DUTAGAM), Düzce, Türkiye

* recaiarslan@duzce.edu.tr

## Abstract

*Ganoderma lucidum* (Curtis) P. Karst (Reishi) has significant pharmacological benefits, and optimizing its cultivation on diverse substrates enhances its commercial viability. This study explored the valorization of various lignocellulosic wastes for cultivating *G. lucidum* mushrooms, highlighting its potential contributions to sustainable agriculture and waste management. In this research, mushrooms were cultivated using hazelnut branches (HB), hazelnut husk (HH), wheat straw (WS), rhododendron branches (RD), oak wood (OW), beech wood (BW), corn husk (CH), and spent coffee grounds (CG). The biodegradation properties of the mushrooms on the selected substrates were also investigated. HB and RD materials were used for the first time to cultivate and assess the biodegradation properties of *G. lucidum*. Substrates were prepared for cultivation at varying compositions (91% substrate + 9% wheat bran (WB) and 75% HB + 25% other substrates). The nutritional properties of the mushrooms and substrates, the chemical composition (holocellulose, alpha cellulose, extractives, and ash) of the substrates, and Fourier Transform Infrared Spectroscopy (FT-IR) assessments before and after cultivation were analyzed. Among the substrates, OW 91% + WB 9% yielded the highest mushroom production (46 g/kg), whereas CH (18.3 g/kg) resulted in the lowest yield, with no significant difference compared to WS (18.5 g/kg). Following mushroom cultivation, the holocellulose content and pH values of the substrates decreased proportionally, while alpha-cellulose, extractives, and ash content increased. Chemical analysis revealed an average holocellulose reduction of 13.5% and α-cellulose increase of 32%, alongside substrate-dependent phenolic content variations, with the highest level (3.156 mg GAE/g) observed in beech wood-grown specimens. FT-IR spectra indicated that this method could effectively elucidate the biodegradation properties of *G. lucidum* on lignocellulosic materials before and after cultivation.

**Data availability statement:** The data underlying the results presented in the study are available from (figshare; https://figshare.com/s/d576ac2e82092913de7b, https://figshare.com/s/fb656666ce7db3a68cdb).

**Funding:** This research was financially supported by Directory of Scientific Research Projects of Düzce University (project number 2024.29.01.1446). The funders had no role in study design, data collection and analysis, decision to publish, or preparation of the manuscript.

**Competing interests:** The authors have declared that no competing interests exist.

## 1. Introduction

*Ganoderma lucidum* (Curtis) P. Karst (Reishi), also known as Ling Chu, Ling Zhi, or Mannetake in China, Korea, and Japan, is a medicinal mushroom that has been widely used in traditional medicine for approximately 2000 years in Eastern countries [1,2]. It is renowned for its pharmacological activities, including antiviral, antibacterial, antifungal, anticancer/cytotoxic, anti-aging, antiallergic, immunomodulating, antidepressant, anti-inflammatory, antioxidative, antihyperlipidemic, antidiabetic, anti-ulcer, hepatoprotective, neuroprotective, nephroprotective, osteoprotective, and hypotensive effects [3,4]. Due to its extensive medicinal properties, *G. lucidum* has been traditionally believed to contribute to longevity. It has been reported to be effective in treating various diseases, such as cancer, hepatitis, allergies, bronchitis, gastric ulcers, arthritis, chronic hepatitis, hyperglycemia, hypertension, insomnia, and inflammation [3].

Today, several commercial products derived from *G. lucidum* are available on the market in various forms, including powder, capsules, tablets, coffee, tea, syrups, beverages, nutritional supplements, lotions, and toothpaste. Thousands of tons of *G. lucidum* are produced and consumed globally, with China being the largest producer, accounting for approximately 70% of the world's total production [5,6]. The global annual production of *G. lucidum* is estimated to be around 4700 tons, with 3700 tons produced in China alone [6].

*Ganoderma lucidum* is classified as a white-rot fungus due to its ability to degrade lignocellulosic materials efficiently. It plays a significant role in the decomposition and recycling of lignocellulosic compounds in nature. Consequently, it is cultivated on various agricultural and forestry residues, which serve as suitable substrates for its growth [2]. Globally, approximately 998 million tons of agricultural waste are generated annually [7]. Lignocellulosic wastes, primarily composed of cellulose, hemicellulose, and lignin, can be utilized as substrates for mushroom cultivation. Despite an estimated $6 \times 10^9$ tons of lignocellulosic waste being produced worldwide each year, only 20% is converted into value-added products such as biofuels, animal feed, or human food (e.g., mushrooms) [8]. Agricultural and lignocellulosic wastes originate from industries such as pulp and paper, food processing, agriculture, and forestry. In many regions, these wastes are often burned after harvesting, contributing to environmental pollution [9]. Various methods have been explored to mitigate this issue, and mushroom cultivation has emerged as one of the most environmentally friendly approaches for converting lignocellulosic materials into high-value products.

A wide range of lignocellulosic materials have been investigated for the cultivation of edible and medicinal *G. lucidum.* Previous studies have evaluated substrates such as tea waste, eggshells, olive oil extraction residues, soybean meal, date palm leaves, wheat bran, banana peels, sugarcane bagasse, and coffee grounds [10–12]. The choice of substrate significantly affects the nutritional composition, including macro- and microelements, and bioactive compounds of the cultivated mushrooms [2]. However, the availability and cost of these lignocellulosic materials vary across different countries and regions [13]. Hazelnut branch pruning waste is generated

during the hazelnut harvest season as a renewable resource, yet it is not widely utilized in industrial applications. In Turkey, local farmers commonly burn these residues for heating. It has been reported that 1.7 million tons of hazelnut branch pruning waste are generated annually from 0.65 million tons of hazelnut production [14,15]. Since hazelnut branches contain high amounts of cellulose, hemicellulose, and lignin, they hold potential as an alternative substrate for mushroom cultivation. While prior studies used tea waste, sugarcane bagasse, or soybean meal [10,12], we focus on regionally abundant wastes (hazelnut branches, rhododendron) to reduce disposal costs and enhance sustainability. Hazelnut branches have been previously used to cultivate oyster mushrooms (*Pleurotus ostreatus* (Jacq.) P. Kumm) [16]. However, no prior research has been conducted on the cultivation and biodegradation properties of *G. lucidum* using hazelnut branches and rhododendron waste as substrates. Here, biodegradation properties refer to *G. lucidum*'s capacity to decompose lignocellulosic components (cellulose, hemicellulose, lignin) quantified via FT-IR and chemical assays. *Rhododendron* sp. branches were selected due to their regional abundance as forestry waste, aligning with our goal to valorize underutilized lignocellulosic materials.

This study aimed to cultivate the medicinal mushroom *G. lucidum* on various lignocellulosic wastes, including hazelnut branches, hazelnut husks, wheat straw, rhododendron branches, oak wood, beech wood, corn husks, and spent coffee grounds. Additionally, the biodegradation properties of *G. lucidum* were assessed through chemical analysis and Fourier Transform Infrared Spectroscopy (FT-IR) after cultivation. The findings of this study provide valuable insights into the efficient utilization of agricultural and forest wastes for *G. lucidum* cultivation, offering a sustainable solution for waste management and the production of high-value medicinal mushroom.

## 2. Material and method

### 2.1. Preparation of mixtures from waste materials

In this study, hazelnut branches (*Corylus avellana L.)* (HB)*,* hazelnut husks (HH), wheat straw (WS), spent coffee grounds (CG), *Rhododendron sp*. (RD), beech wood (*Fagus orientalis* Lipsky) (BW), oak wood (*Quercus sp.)* (OW)*,* and corn husk (CH) wastes were used as substrates for *G. lucidum* cultivation. Hazelnut branches, hazelnut husks, corn husks, *Rhododendron sp.*, and wheat straw were obtained from local growers, while beech wood and oak wood sawdust were sourced from wood processing factories. Spent coffee grounds were collected from a local coffee company in Düzce, Turkey.

The substrates were processed into suitable sizes: HB and HH were ground using a Wiley mill to 1 cm, *Rhododendron sp.* was cut to 2 cm, WS was prepared at a size of 5–6 cm, and CG was sieved to 0.3–0.5 mm. Beech and oak wood sawdust were sieved to 1.4 mm before use.

A total of 15 different substrate mixtures were prepared using lignocellulosic materials in homogeneous weight ratios (w/w) (Table 1). The substrate proportions were carefully selected based on established cultivation protocols and waste valorization objectives. Control groups utilized a standardized substrate composed of 91% substrate + 9% wheat bran (containing of 1.88% N content) [10], with wheat bran serving as an optimal nitrogen supplement to balance the C/N ratio. For hazelnut branch (HB) mixtures, a 75% HB + 25% other materials ratio was implemented to specifically address two key considerations: the material's significance as Turkey's prominent agricultural waste (1.7 million tons/year; [16]) with favorable lignocellulosic composition for fungal growth, and (2) the need to evaluate potential synergistic effects of substrate combinations while preserving HB's structural integrity. This experimental design enabled systematic comparison between conventional substrates and novel waste-based alternatives.

Substrates were initially weighed as dry materials (500 g per bag, except for high-volume CH and WS which were reduced to 250 g to accommodate bag capacity). The mixtures were wetted until their moisture content reached approximately 65% [10]. Each combination was weighed to 1 kg, and 1% $CaCO_3$ was added to adjust the pH. The mixtures were then placed into autoclavable polypropylene bags (28 × 42 cm) and sterilized in an autoclave at 121°C and 1.1 atm pressure for 90 minutes [2]. At least six replicates were prepared for each combination. After sterilization, the pH and moisture content of each mixture were measured, and the substrates were allowed to cool to 24°C before inoculation.

**Table 1. Substrate codes and ratios used in the study.**

| Substrates | Substrate code | Substrate ratio % |
|---|---|---|
| Hazelnut Branch + Wheat Bran | HB + WB | 91 + 9 |
| Rhododendron + Wheat Bran | RD + WB | 91 + 9 |
| Beech Wood + Wheat Bran | BW + WB | 91 + 9 |
| Oak Wood + Wheat Bran | OW + WB | 91 + 9 |
| Hazelnut Husk + Wheat Bran | HH + WB | 91 + 9 |
| Coffee Ground + Wheat Bran | CG + WB | 91 + 9 |
| Wheat Straw + Wheat Bran | WS + WB | 91 + 9 |
| Corn Husk + Wheat Bran | CH + WB | 91 + 9 |
| Hazelnut Branch+ Hazelnut Husk | HB + HH | 75 + 25 |
| Hazelnut Branch+ Rhododendron | HB + RD | 75 + 25 |
| Hazelnut Branch + Wheat Straw | HB + WS | 75 + 25 |
| Hazelnut Branch + Beech Wood | HB + BW | 75 + 25 |
| Hazelnut Branch + Oak Wood | HB + OW | 75 + 25 |
| Hazelnut Branch + Coffee Ground | HB + CG | 75 + 25 |
| Hazelnut Branch + Corn Husk | HB + CH | 75 + 25 |

The *G. lucidum* spawn used in the study was obtained from the Atatürk Horticultural Central Research Institute, Yalova, Turkey. The spawn was inoculated into the substrate at a rate of 3% (w/w) of the dry compost weight inside a biosafety cabinet. The inoculated composts were mixed homogeneity, tightly sealed, and incubated in a dark environment.

## 2.2. Incubation and mushroom harvesting

The inoculated substrates were transferred to a dark incubation chamber maintained at 24 ± 2°C and 70% relative humidity [10]. The inoculated substrates were incubated until complete mycelial colonization was achieved. Colonization progress was monitored daily through transparent polypropylene bags. Once full mycelial colonization was observed, the incubation temperature was increased to 28–30°C [12], and the humidity was adjusted to 80–90%. Carbon dioxide ($CO_2$) levels in the chamber were maintained below 1000 ppm [10].

All cultivation bags remained hermetically sealed during the entire process, with only the sterile cotton filter at the neck being removed to permit the directional growth and emergence of fruiting bodies. When primordium/pin formation appeared on the mycelium, 50–60 lux light per m² was introduced to the incubation chamber to stimulate fruit body development. The growth chamber was illuminated by standard ceiling-mounted fluorescent/LED fixtures providing cool white spectrum and ~5000K suitable for *G. lucidum* primordiation. Light cycles were maintained at 10–12 hour photoperiods through manual operation [2,10]. Fully matured fruiting bodies were harvested and weighed using a precision scale.

A total of three harvesting cycles were performed for each substrate combination. Spawn run time was recorded as the duration for complete mycelial colonization, assessed via daily visual inspection. Earliness represented the period from inoculation to first harvest-ready fruiting body. Dry matter content was determined by oven-drying fresh mushrooms at 40°C to constant weight.

The mushroom yield and biological efficiency (BE) were calculated using the following formulas:

$$\text{Yield (g/kg)} = (\text{Total weight of fresh mushrooms harvested g/1 kg substrate}) \quad (1)$$

$$\text{BE (\%)} = (\text{Total weight of fresh mushrooms harvested/dry matter content of the substrate}) \times 100 \quad (2)$$

## 2.3. Chemical analysis of the substrates

**2.3.1. Extractive, holocellulose, alpha-cellulose, ash content.** The extractives content of both pre-cultivation control substrates (HB-C, HH-C, WS-C, RD-C, OW-C, BW-C, CH-C, CG-C) and post-cultivation fungal-degraded materials (HB-F, HH-F, WS-F, RD-F, OW-F, BW-F, CH-F, CG-F) was determined according to a modified TAPPI T 204 cm-17 standard. All substrate samples were first ground to 40-mesh particle size using a Wiley mill and then oven-dried at $103 \pm 2°C$ until constant weight was achieved. For the extraction process, precisely weighed 5 g aliquots of each dried sample were subjected to solvent extraction (toluene: acetone: ethanol mixture [4/1/1, (v/v)] in a Soxhlet apparatus for 6 hours, with the extraction cycles carefully monitored to ensure complete removal of soluble compounds. Following extraction, the solutions were vacuum-filtered through pre-weighed porosity-2 crucibles, and the retained residues were subsequently dried at $103 \pm 2°C$ for 12 hours. The extractives content was calculated gravimetrically by comparing the initial sample mass to the mass of insoluble residue after extraction, with all measurements performed in triplicate to ensure reproducibility [17].

The holocellulose content of the eight substrate samples before and after cultivation (hazelnut branches, hazelnut husk, corn husk, coffee grounds, rhododendron branches, oak wood, beech wood, and wheat straw) was determined using an optimized Wise and John chlorite method. Extractives-free 40-mesh samples (5 g), prepared by oven-drying at $103 \pm 2°C$, were subjected to a four-stage delignification process in 250 mL Erlenmeyer flasks. Each reaction cycle consisted of adding 160 mL distilled water, 1.5 g sodium chlorite ($NaClO_2$), and 0.5 mL glacial acetic acid, followed by incubation at $78 \pm 1°C$ in a temperature-controlled water bath with continuous magnetic stirring. The mixture underwent vigorous shaking every 15 minutes during the 1-hour reaction period to ensure homogeneous treatment. After each cycle, fresh reagents (1.5 g $NaClO_2 + 0.5$ mL acetic acid) were added, with the complete process repeating four times to achieve thorough lignin removal. The resulting holocellulose was collected by vacuum filtration through pre-weighed porosity-2 glass crucibles, then sequentially washed with ice-cold distilled water (to remove residual acids) and acetone (to facilitate drying). The purified holocellulose was dried to constant weight at $103 \pm 2°C$. The holocellulose content of the substrates (%) were determined relative to the initial full dry weight [18].

Alpha-cellulose content was determined following TAPPI T 203 cm-09 with modifications. Precisely 2 g of holocellulose was treated with 17.5% NaOH (10 mL initial + 2 × 5 mL additions at 5-min intervals) at $20.0 \pm 0.5°C$ for 30 min. After dilution with 33 mL distilled water and 60 min incubation, the mixture was filtered through porosity-2 crucibles. The residue was sequentially washed with 8.3% NaOH (100 mL), 10% acetic acid (15 mL), and distilled water (250 mL), then dried at 105°C to constant weight. Alpha-cellulose content was calculated gravimetrically relative to initial holocellulose mass, with triplicate measurements [19].

Ash content was calculated by combusting substrates at 575°C until constant weight (TAPPI T211 om-16) to determine inorganic residues. pH was measured in a 1:10 (w/v) substrate-water suspension using a calibrated pH meter [20]. Comparative analyses were performed on each substrate type in both pre-cultivation (raw) and post-cultivation (spent) states to quantify fungal degradation effects.

**2.3.2. Total organic carbon, Nitrogen and C/N.** Total organic carbon (TOC) content was quantified using dry combustion at 680°C with a TOC analyzer (Elementar soli TOC). The ground substrate samples obtained from the raw materials before cultivation (50 mg) were combusted in crucibles, and evolved $CO_2$ was measured via nondispersive infrared detection. Total nitrogen content was determined using the Kjeldahl method (AOAC 978.02), involving sulfuric acid digestion with a copper sulfate catalyst, followed by distillation and titration of liberated ammonia. The carbon-to-nitrogen (C/N) ratio was calculated as the quotient of TOC and total nitrogen percentages.

## 2.4. Proximate composition of fruiting bodies

**2.4.1. Total phenolic contents.** The total phenolic content (TPC) of *G. lucidum* fruit bodies was determined using the Folin-Ciocalteu method [21]. First, 0.5 g of dried mushroom powder was mixed with 10 mL methanol and filtered. Then,

800 μL Folin-Ciocalteu reagent and 1 mL $Na_2CO_3$ solution were added. The solution was incubated in the dark for 30 minutes, and absorbance values were recorded at 720 nm using a spectrophotometer. The results were expressed as mg gallic acid equivalent per gram (mg GAE/g) of the sample.

**2.4.2. Elemental analysis.** Elemental analysis was performed using Inductively Coupled Plasma Optical Emission Spectrometry (ICP-OES), following the ISO 22036 [22] standard protocol for environmental solid matrices. Certified reference materials (Certipur, Merck, Germany) were used for ICP-OES calibration. Prior to analysis, mushroom compost and fruit body samples were digested in a microwave oven using a mixture of nitric acid ($HNO_3$) and hydrogen peroxide ($H_2O_2$).

## 2.5. FT-IR measurements

FT-IR spectra were obtained using an IRPrestige-21 FT-IR Spectrophotometer (Shimadzu) equipped with a single-reflection ATR sampling module. The spectra were recorded in the range of 4000–400 cm$^{-1}$ with a resolution of 4 cm$^{-1}$ by averaging 32 scans. Spectral data in the region between 1800 and 800 cm$^{-1}$ were processed using curve fitting analysis performed in the OriginPro 2013 software (OriginLab Corporation, Northampton, MA 01060 USA), by determining the area under each band. The FT-IR spectra were fitted using Gaussian-shaped bands. Optimal Gaussian curve fitting was determined when the best fit was achieved (reduced chi-square $< 1 \times 10^{-6}$) and based on the agreement between the calculated areas (R-square, $R^2 = 0.998–0.987$ for substrates, $R^2 = 0.901–0.941$ for mushrooms) (see S1-S22 in S1 File for reference).

## 2.6. Statistical analysis

Statistical analyses were carried out using SPSS 19 software (IBM Corp., Armonk, NY, USA). The averages were evaluated using a one-way analysis of variance (ANOVA). Duncan's multiple range test was applied at the level of $\alpha = 0.005$ to determine the significant differences among the paired comparisons (compost contents and mushroom quality properties).

## 3. Results and discussion

### 3.1. Spawn run time, earliness, yield, biological efficiency and dry matter

Table 2 presents the spawn run time, earliness (days to first harvest), yield, biological efficiency (BE), and dry matter content of mushrooms cultivated on different substrates. All the tested substrates were successfully colonized by *G. lucidum* mycelium; however, significant differences ($p < 0.05$) were observed in the spawn run time across the different substrates. The shortest spawn run time (5 days) was recorded in the RD 91% + WB 9% mixture, while the longest colonization period (14 days) was observed in HH 91% + WB 9% and CH 91% + WB 9% substrates.

Atila et al [2] used oak wood (OW) as a basal substrate or control for the cultivation of *G. lucidum*. The spawn run time for OW was found to be 7 days in the current study, which was shorter than the spawn run time (18 days) reported. The mycelium used, the species of oak wood, and the environmental conditions can be considered factors affecting the spawn run time. The longer spawn run time in HH 91% + WB 9% substrates may be related to the high nitrogen or lignin content in the HH material [16].

Significant differences were found between mean earliness values across substrates ($p < 0.05$). The lowest earliness (43 days) was detected in the OW 91% + WB 9% substrate. The earliness value for OW was much shorter than the reported values by Atila [2], Gurung [23], and Roy [24]. The shorter earliness in OW may reflect its optimal C/N ratio (50.68) and lignin content (~25%), which enhance mycelial efficiency (See Table 3) [28]. HB material was used for the first time in this study for cultivation with various agricultural wastes, and its mean earliness was found to be 47.2 days, which was not significantly different from OW (43 days), RD (43.5 days), and BW (51 days). When other agricultural wastes were supplemented with 25% to the HB, the mean earliness did not significantly differ ($p > 0.05$). Although all substrates were colonized with mycelium, primordia and fruit bodies did not occur in the HH 91% + WB 9%, CG 91% + WB 9%, and HB 75% + CG 25% substrates.

**Table 2. Spawn run time, earliness, yield, biological efficiency and dry matter of mushroom fruit body (see S1 in S2 File for reference).**

| Substrates | Spawn run time (days) | First promordium (days) | Earliness (days) | Yield g/kg | BE (%) | Dry matter % |
|---|---|---|---|---|---|---|
| HB 91%+9% WB | 7[b] | 21.7[abc] | 47.2[ab] | 28.3[bc] | 5.65[ab] | 28.2[bc] |
| RD 91%+9% WB | 5[a] | 21[ab] | 43.5[a] | 36.3[c] | 7.25[ab] | 27.0[b] |
| BW 91%+9% WB | 11[c] | 23.42[cd] | 51[bc] | 27[b] | 5.39[a] | 26.4[b] |
| OW 91%+9% WB | 7[b] | 21.1[ab] | 43[a] | 46[d] | 9.20[c] | 22.6[a] |
| HH 91%+9% WB | 14[d] | * | * | * | * | * |
| CG 91%+9% WB | 7[b] | * | * | * | * | * |
| WS 91%+9% WB | 11[c] | 23.6[cd] | 53[cd] | 18.8[a] | 7.54[bc] | 27.10[b] |
| CH 91%+9% WB | 14[d] | 24.6[e] | 55.1[d] | 18.5[a] | 7.4[bc] | 32.03[de] |
| HB 75%+HH 25% | 7[b] | 24.1[e] | 49.4[bc] | 29.4[bc] | 5.88[ab] | 32.09[e] |
| HB 75%+RD 25% | 7[b] | 20.4[a] | 44.8[a] | 33.4[bc] | 6.68[ab] | 26.56[b] |
| HB 75%+WS 25% | 8[c] | 20.5[a] | 49.1[bc] | 29.2[bc] | 5.85[ab] | 29.64[bcd] |
| HB 75%+BW 25% | 7[b] | 23.1[bcd] | 51.8[cd] | 34.3[bc] | 6.86[ab] | 30.88[cde] |
| HB 75%+OW 25% | 7[b] | 20.8[ab] | 49.8[bc] | 28.5[bc] | 5.70[ab] | 29.41[bcd] |
| HB 75%+CG 25% | 7[b] | * | * | * | * | * |
| HB 75%+CH 25% | 7[b] | 23.1[bcd] | 52.8[cd] | 34.0[bc] | 6.81[ab] | 29.20[bcd] |

* Mushroom fruit body was not occurred.

**Table 3. Total organic carbon and phenolic content of raw materials and fruit body cultivated (see S2 in S2 File for reference).**

| Substrate type | Average TOC (%) | N | C/N | Total phenolic content (mg GAE/g sample) | |
|---|---|---|---|---|---|
| | Substrates | Substrates | Substrates | Substrates | Fruit body |
| **HB** | 20.02±0.110[ab] | 1.23±0.07[a] | 16,27±0.97[ab] | 1.751±0.001[a] | 2.191±0.001[a] |
| **RD** | 21.82±0.445[d] | 0.5±0.08[b] | 43,64±4.1[d] | 14.872±0.001[b] | 2.619±0.000[b] |
| **BW** | 19.77±0.790[ab] | 0.44±0.04[b] | 44,93±9.8[d] | 14.083±0.003[c] | 3.156±0.005[c] |
| **OW** | 20.78±0.160[c] | 0.41±0.08[b] | 50,68±5.4[d] | 2.083±0.001[d] | 2.689±0.000[d] |
| **HH** | 19.40±0.065[a] | 1.51±0.08[c] | 12,84±0.66[a] | 4.423±0.005[e] | * |
| **CG** | 21.69±0.145[d] | 2.52±0.11[d] | 8,60±0.35[a] | 3.199±0.001[f] | * |
| **WS** | 20.00±0.510[ab] | 0.68±0.06[e] | 29,41±4.5[c] | 2.513±0.001[g] | 1.687±0.001[e] |
| **CH** | 20.20±0.375[bc] | 0.83±0.06[f] | 24,33±1.6[bc] | 1.787±0.001[h] | 2.909±0.005[f] |

* Mushroom fruit body was not occurred.

When yield values were examined, the highest yield (46 g/kg) was achieved in the substrate prepared with OW 91%+WB 9% (significantly different from the other substrates, $p < 0.05$), while the lowest yield was found in the substrates prepared with WS 91%+WB 9% and CH 91%+WB 9%, at 18.8 and 18.5 g/kg, respectively. In *G. lucidum* cultivation, beech wood (BW) and oak wood (OW) are widely recognized as the traditional basal substrate due to its optimal lignin content and C/N ratios (approximately 45~50) which are favorable fruiting body formation. Compared to alternative substrates like wheat straw (WS), OW demonstrated superior performance in this study, yielding 46 g/kg—a 148% increase over WS (18.5 g/kg). Since no fruit body formation occurred in the substrates prepared with HH 91%+WB 9%, CG 91%+WB 9%, and HB 75%+CG 25%, the yield value and BE were not calculated. HH and CG materials were reported by Akcay et al [16] and Ozcelik and Peksen [25] as suitable cultivation materials for *Pleurotus ostreatus* (Oyster mushroom) and *Lentinus edodes* (Shiitake), both of which are edible and medicinal mushroom species. However, some studies have reported that increasing the level of spent coffee grounds in sawdust substrates delayed mycelium growth and

prevented fruiting during mushroom cultivation [26]. A similar issue occurred in the cultivation of *G. lucidum* in the current study. The yield of the substrate prepared with HB material, which was used for the first time in *G. lucidum* cultivation, was 28.3 g/kg. The RD material, also used for the first time, had a yield of 36.3 g/kg. When HB and RD materials were mixed with other lignocellulosic materials at a ratio of 25%, the yield values were not significantly different (p < 0.05). It seems that the supplementation of lignocellulosic materials (at 25%) to HB did not negatively affect the yield. When comparing our results with previous studies, the yield determined for OW 91% + WB 9% (46 g/kg) in the current study was comparable with the results reported by Yakupoglu [27] and Peksen and Yakupoglu [10], who investigated the yields of substrates prepared by mixing sawdust and wood chips of oak and hornbeam with tea wastes (42 g/kg). However, the yield values of other substrates were lower than those in their results across all combinations. The differences and lower yields could be attributed to variations in substrate combinations, pH values, and cultivation conditions. BE values showed similar trends to yield values. The highest BE value (9.20%) was found in substrates prepared with OW 91% + WB 9%, while the lowest (5.39%) was found in the BW 91% + WB 9% substrate.

Significant differences were detected between substrates in terms of dry matter in mushroom fruit bodies (p < 0.05). The highest dry matter content was found to be 28.2%, while the lowest was 22.6%, among those grown on different lignocellulosic materials at a ratio of 91% and WB 9%. When dry matter values from the HB 75% and different lignocellulosic material substrates with 25% were examined, the highest dry matter (32.09%) was achieved in the HB 75% + HH 25% substrate. Dry matter content in *G. lucidum* mushroom fruit bodies is an important characteristic, as it is valued when dried in the market [5].

### 3.2. Total organic carbon (TOC), Nitrogen, C/N, Phenolic content (TPC) and Elemental composition

The total organic carbon (TOC), nitrogen (N), C/N ratio, and phenolic content of the substrates, as well as the phenolic content of the cultivated mushrooms, are shown in Table 3. When examining Table 3, the highest TOC was found in RD at 21.82%, while the lowest was in HH at 19.4%. However, TOC values were generally comparable. When the nitrogen contents of the substrates were analyzed, the highest N content was observed in CG (2.52%), followed by HH (1.51%). The lowest nitrogen content was found in OW and BW (0.41% and 0.44%, respectively). The C/N ratio is a critically important indicator in mushroom production. In this study, the lowest C/N ratios were found in CG and HH substrates. It can be suggested that the low C/N ratio in these substrates plays an active role in the formation of mushroom caps. The highest C/N ratio was observed in the OW substrate at 50.68%, followed by RD and BW. Yield values were also high in substrates with high C/N ratios. Similar results were reported by Peksen and Yakupoglu [10], who found a strong positive correlation between the C/N ratio of substrates and mushroom yield. Atila [28] also found that yield was strongly positively correlated with C/N ratio, cellulose, and lignin content, while it was negatively correlated with the nitrogen content of the substrates.

The relationship between substrate C/N ratios and *G. lucidum* productivity is complex and substrate-dependent, though a clear positive correlation emerges across studies. While agricultural waste-based substrates like wheat straw and tea waste achieve peak yields at higher C/N ratios (70–80) due to their lignin-rich composition requiring extended carbon availability for degradation [10], wood-based substrates such as oak perform optimally at moderate ratios (~50) where nitrogen availability better supports mycelial expansion [24]. Our results align precisely with these mechanisms: low C/N substrates (e.g., coffee grounds at 8.6) failed to fruit entirely, while oak wood (C/N 50.7) yielded maximally (46 g/kg), corroborating Atila's [28].) findings of yield being positively correlated with C/N (r = 0.82, p < 0.01) and negatively with nitrogen content. This pattern reflects *G. lucidum*'s dual metabolic demands – sufficient nitrogen for biomass accumulation balanced against carbon reserves for ligninolytic enzyme production during fruiting. Consequently, ideal C/N ranges must account for both substrate type (agricultural vs. woody) and cultivation phase (colonization vs. fruiting), with our 25–50 range accommodating this plasticity.

When the total phenolic content (TPC) was examined, the highest TPC among the substrates was found in RD, with a value of 14.872 mg GAE/g. The highest TPC in mushroom fruit bodies was recorded in BW at 3.156 mg GAE/g, while the

lowest TPC among mushroom fruit bodies was found in mushrooms cultivated on WS, with a value of 1.687 mg GAE/g (~87% greater than those from WS). Modi et al [29] reported that the TPC values of *G. lucidum* extracts collected from nature varied between 33.833 and 83.674 mg/g dry extract, depending on the solvent used in their study. Atila et al [2] found TPC values ranging from 20.36 to 22.60 mg GAE/g in *G. lucidum* mushrooms cultivated on various phenolic-rich forest and agro-food waste substrates. Similarly, Demirci et al [30] reported that the TPC values of five different commercially available *G. lucidum* samples ranged between 2.35 and 10.46 mg GAE/g. It appears that the TPC values of the mushrooms in this study were considerably lower than those reported by Modi et al [29] and Atila et al [2], but comparable to the values reported by Demirci et al [30]. This discrepancy can be attributed to several interconnected factors that collectively influence phenolic compound production and extraction efficiency. One of them is that the agricultural waste substrates used in our cultivation system appear to have preferentially stimulated polysaccharide biosynthesis over phenolic compound production, contrasting with the metabolic patterns observed in forest-grown specimens developing on natural woody substrates [2]. Another factor maybe related with the commercial harvesting protocol we followed, targeting optimal fruiting body size rather than peak metabolite accumulation, likely captured a different developmental stage than the mature specimens typically collected from wild populations [4]. Additionally, the domesticated strain utilized in this study, obtained from the Atatürk Horticultural Institute, may have been selectively bred for rapid growth characteristics rather than secondary metabolite production [6]. While Demirci et al. [30] have clearly established the significant impact of extraction solvents on measured TPC values, our findings further highlight how cultivation parameters – including substrate composition, genetic strain characteristics, and harvest timing – serve as equally critical determinants of phytochemical profiles.

The elemental analysis of substrates and cultivated mushroom fruit bodies in this study is presented in Table 4. Upon examining Table 4, significant differences were observed among the substrates in terms of mineral composition. Essential minerals such as Fe, Cu, Zn, Mn, Mg, and Ca are required for fungal growth. Peksen and Yakupoglu [10] found that yield

**Table 4. Elemental analysis of substrates and fruit bodies cultivated.**

| Sample | N (wt%) | P (wt%) | Fe (mg/kg) | Cu (mg/kg) | Zn (mg/kg) | Mn (mg/kg) | Na (mg/kg) | Mg (mg/kg) | K (mg/kg) | Ca (mg/kg) |
|---|---|---|---|---|---|---|---|---|---|---|
| **Substrates** | | | | | | | | | | |
| HB | 1.23 | 0.1 | 80.4 | 38.6 | 33.3 | 28.6 | 159.1 | 2006.0 | 8531.8 | 1468.9 |
| RD | 0.5 | 0.0 | 87.2 | 31.9 | 41.4 | 133.6 | 229.5 | 1382.7 | 12575.0 | 1570.0 |
| BW | 0.44 | 0.0 | 217.8 | 57.7 | 4.2 | 88.2 | 239.5 | 1649.9 | 10282.1 | 2470.0 |
| OW | 0.41 | 0.0 | 117.2 | 299.1 | 2.4 | 48.9 | 133.6 | 554.3 | 1043.0 | 2896.3 |
| HH | 1.51 | 0.1 | 145.8 | 10.6 | 43.7 | 65.4 | 172.1 | 884.6 | 7917.6 | 971.2 |
| CG | 2.52 | 0.1 | 150.1 | 55.6 | 3.5 | 19.5 | 167.0 | 746.9 | 5128.8 | 5479.6 |
| WS | 0.68 | 0.0 | 95.6 | 7.6 | 95.7 | 155.3 | 232.5 | 1269.4 | 14708.5 | 616.6 |
| CH | 0.83 | 0.1 | 186.0 | 10.8 | 74.6 | 138.6 | 359.5 | 1647.4 | 19819.3 | 782.3 |
| **Fruitbodies** | | | | | | | | | | |
| HB | 2.9 | 0.4 | 148.6 | 35.5 | 3.0 | 31.2 | 119.8 | 116.9 | 996.5 | 6106.7 |
| RD | 3.77 | 0.5 | 90.7 | 264.0 | 13.3 | 36.8 | 274.5 | 914.6 | 9098.1 | 5385.9 |
| BW | 3.71 | 0.3 | 101.2 | 15.1 | 68.3 | 131.1 | 277.7 | 1272.9 | 13907.3 | 993.4 |
| OW | 3.58 | 0.4 | 1351.2 | 1207.2 | 14.1 | 65.9 | 246.9 | 1953.0 | 22297.6 | 12172.6 |
| HH | | | | | | | | | | |
| CG | | | | | | | | | | |
| WS | 3.92 | 0.5 | 91.3 | 79.9 | 2.0 | 38.4 | 202.8 | 457.5 | 1328.4 | 4992.0 |
| CH | 3.5 | 0.5 | 153.4 | 14.6 | 91.9 | 168.6 | 300.8 | 1412.9 | 20094.3 | 735.5 |

* Mushroom fruit body was not occurred.

and biological efficiency (BE) were positively and significantly correlated with K, Fe, and Mn in *G. lucidum* production. Similarly, Ozcelik and Peksen [25] reported that the P and Zn content of substrates was strongly correlated with yield and BE in *Lentinus edodes* (Berk.) Pegler cultivation. When the mineral contents of the cultivated mushroom fruit bodies were analyzed, significant differences were observed. The levels of Fe, Cu, Mg, K, and Ca were higher in OW-FB fruit bodies compared to those grown on other substrates. On the other hand, Zn, Mn, and Na levels were higher in CH-FB fruit bodies than in others.

### 3.3. FT-IR spectroscopy and biodegradation properties

The FT-IR spectra in the fingerprint region (1800–800 cm⁻¹), which show the structural changes in hazelnut branches (HB), beech wood (BW), oak wood (OW), rhododendron branches (RD), corn husk (CH), wheat straw (WS), hazelnut husk (HH), and spent coffee grounds (CG) after fungal attack compared to the control samples (before fungal attack), along with the histograms of areas obtained through the Gaussian curve fitting procedure applied to the examined regions of these spectra, are given in Figs 1–8.

The Gaussian curve fitting procedure applied to the FT-IR spectra of the substrates gave twelve Gaussian curves centered at 890, 1030, 1110, 1150, 1240, 1320, 1370, 1420, 1460, 1510, 1620, and 1730 cm⁻¹. The percentage area of each band showed significant changes after fungal attack. The functional groups corresponding to these bands were examined in detail.

The bands observed between 1750 and 1720 cm⁻¹ are attributed to hemicellulose degradation in lignocellulosic materials. This band indicates unconjugated C=O vibrations of acetyl, carboxylic acid, and uronic ester groups within the hemicellulose structure [31,32]. Hemicellulose degradation caused by fungal attack resulted in a decrease in the area of this band (12) in all samples. The strong band observed at 1742 cm⁻¹ in the CG-C sample is attributed to the C=O stretching vibrations of ester groups from kinic acid and lipids [33]. After the fungal attack, this band disappeared.

The bands between 1650 and 1600 cm⁻¹ correspond to lignin degradation in lignocellulosic materials [31]. In this region, the deformation of O-H groups in the lignin structure (1640 cm⁻¹), C=C and C=O stretching in the lignin aromatic chain (1630 cm⁻¹), and C-O stretching vibrations in the lignin skeletal structure (1630 cm⁻¹) overlapped, forming a broad band

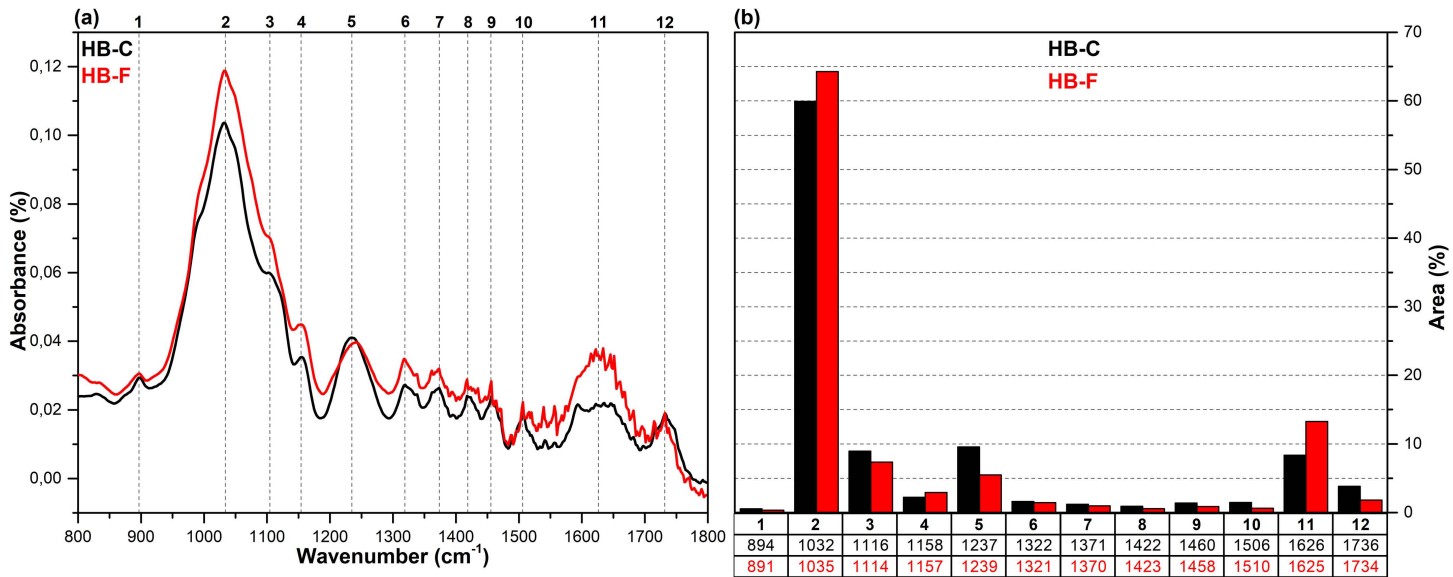

**Fig 1. FT-IR results. a)** FT-IR spectra of HB-C and HB-F; **b)** histograms of FT-IR peak areas processed using Gaussian curve fitting.

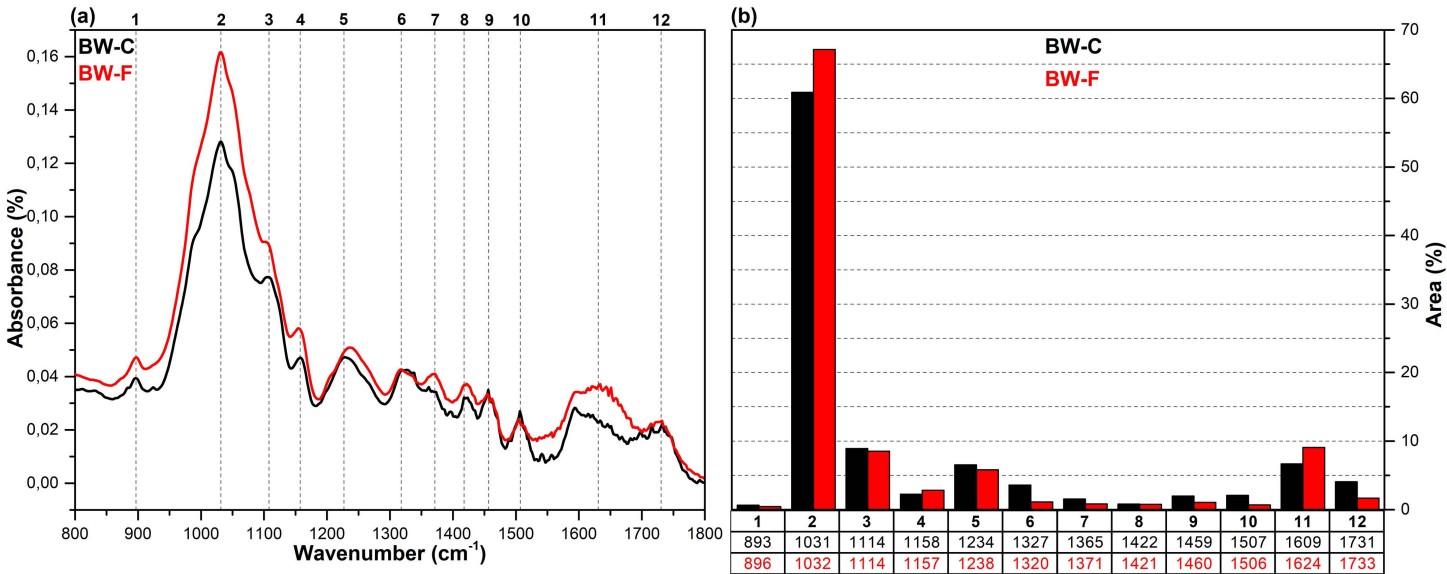

**Fig 2. FT-IR results. a)** FT-IR spectra of BW-C and BW-F; b) histograms of FT-IR peak areas processed using Gaussian curve fitting.

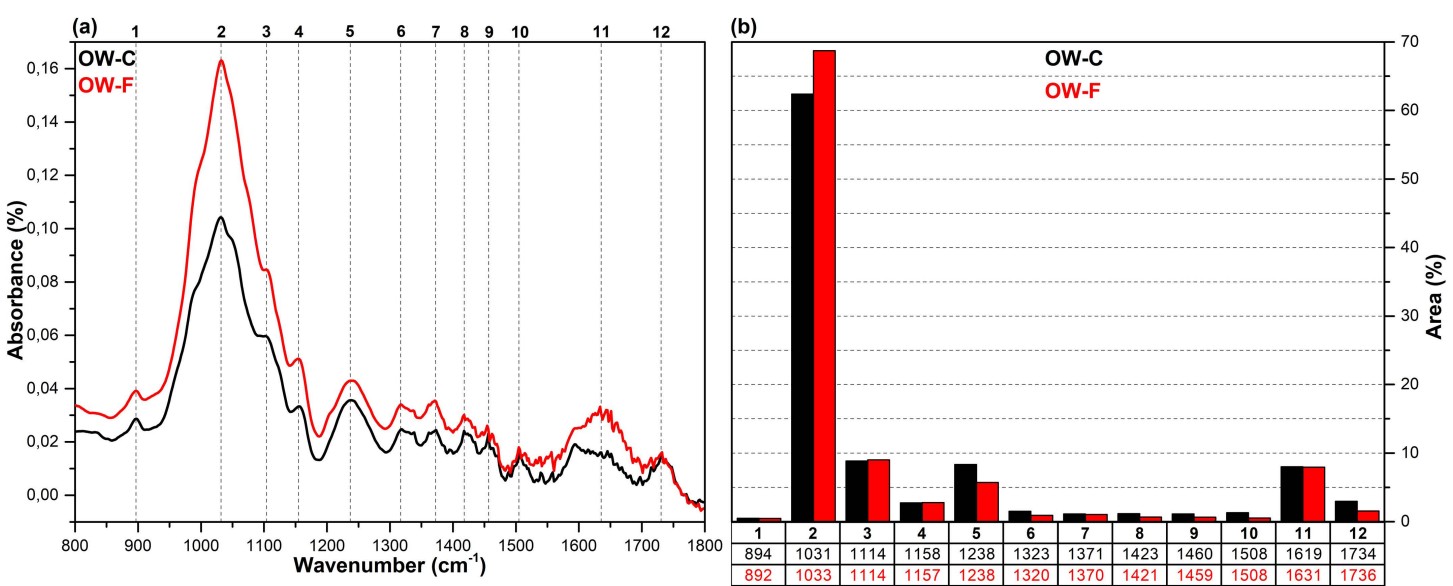

**Fig 3. FT-IR results. a)** FT-IR spectra of OW-C and OW-F; b) histograms of FT-IR peak areas processed using Gaussian curve fitting.

[16]. As a result of lignin degradation by *G. lucidum*, a significant increase in the area of this band (11) was observed in the HB, BW, CH, HH, and CG samples, whereas no notable change was detected in the OW, RD, and WS samples. This result may be associated with the selective lignin degradation by *G. lucidum* mycelia, depending on the type and structure of the substrate.

The broad band observed at $1645\,cm^{-1}$ in the CG-C sample is attributed to C=C stretching vibrations of lipids and fatty acids, C=O stretching vibrations of caffeine, and C=C, C=O, and C-O stretching vibrations of the lignin aromatic chain [34].

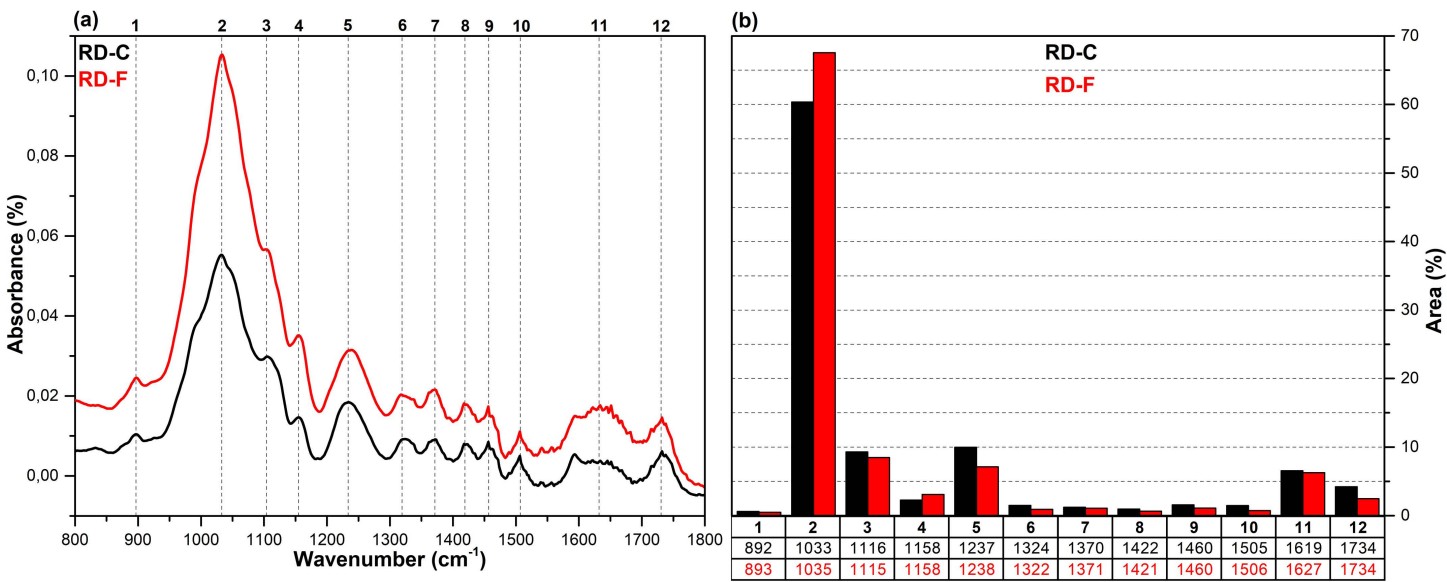

**Fig 4. FT-IR results. a)** FT-IR spectra of RD-C and RD-F; b) histograms of FT-IR peak areas processed using Gaussian curve fitting.

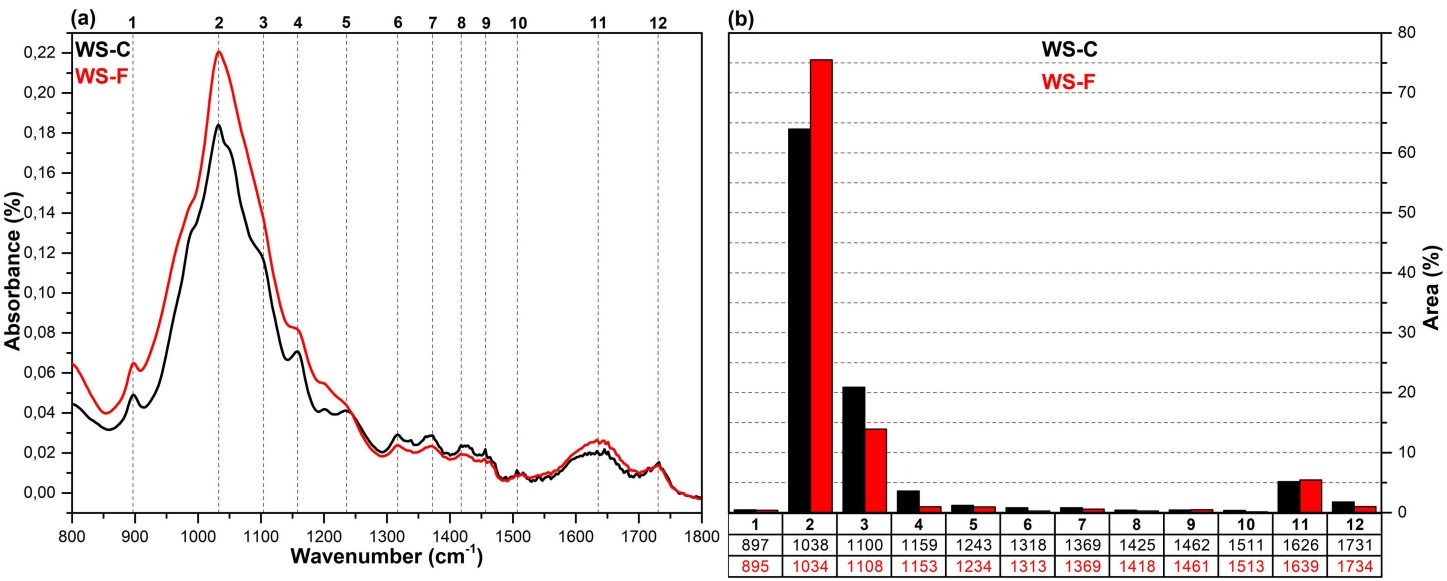

**Fig 5. FT-IR results. a)** FT-IR spectra of WS-C and WS-F; b) histograms of FT-IR peak areas processed using Gaussian curve fitting.

The bands between 1530 and 1500 cm$^{-1}$ correspond to the aromatic C-O stretching vibrations of lignin [35]. Following a fungal attack, the area of this band (10) increased in the CG sample, while it decreased in all other samples. Similarly, Akcay et al [16] reported that *Pleurotus ostreatus* increased the visibility of the 1652 cm$^{-1}$ band due to lignin degradation in coffee. However, in the present study, since fungal growth could not be achieved on the CG substrate, this may be associated with the lower impact of *G. lucidum* on lignin in spent coffee grounds compared to *Pleurotus ostreatus*. This suggests that *G. lucidum* has a weaker effect on lignin in coffee grounds compared to *Pleurotus ostreatus*.

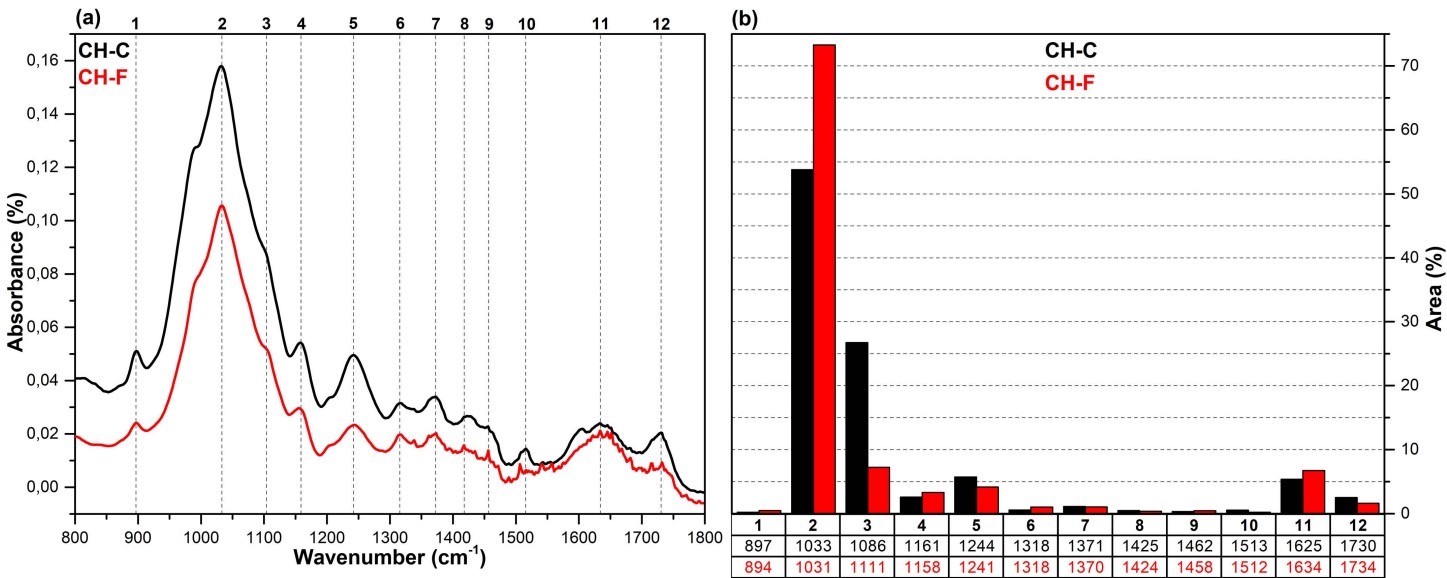

**Fig 6. FT-IR results. a)** FT-IR spectra of CH-C and CH-F; **b)** histograms of FT-IR peak areas processed using Gaussian curve fitting.

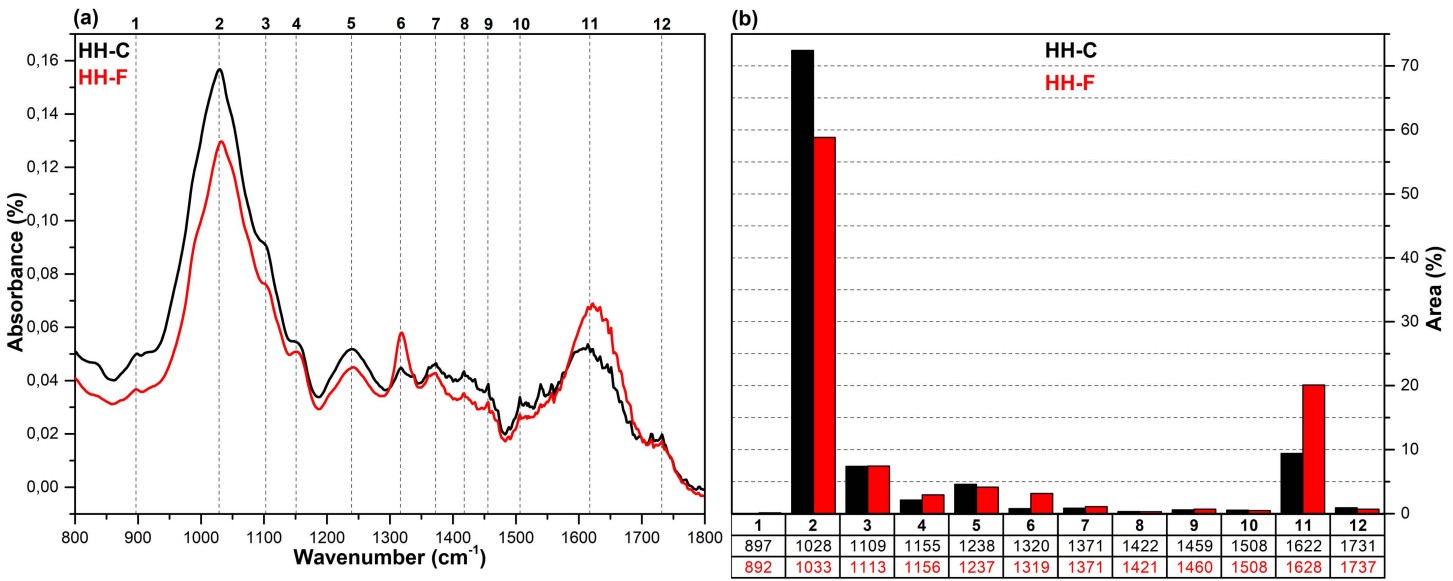

**Fig 7. FT-IR results. a)** FT-IR spectra of HH-C and HH-F; **b)** histograms of FT-IR peak areas processed using Gaussian curve fitting.

The bands between 1465–1455 cm$^{-1}$ and 1425–1410 cm$^{-1}$ are attributed to C-H deformations of CH$_2$ and CH$_3$ groups in lignin and hemicellulose, as well as chlorogenic acids in coffee, and CH$_2$ in-plane bending vibrations in cellulose and lignin, respectively [36–38]. After the fungal attack, the area of the first band (9) decreased in the HB, BW, OW, RD, and CG samples, while it slightly increased in the WS, CH, and HH samples. The area of the second band (8), however, decreased in all samples. A similar result was observed for CG in *Pleurotus ostreatus* mycelia [16].

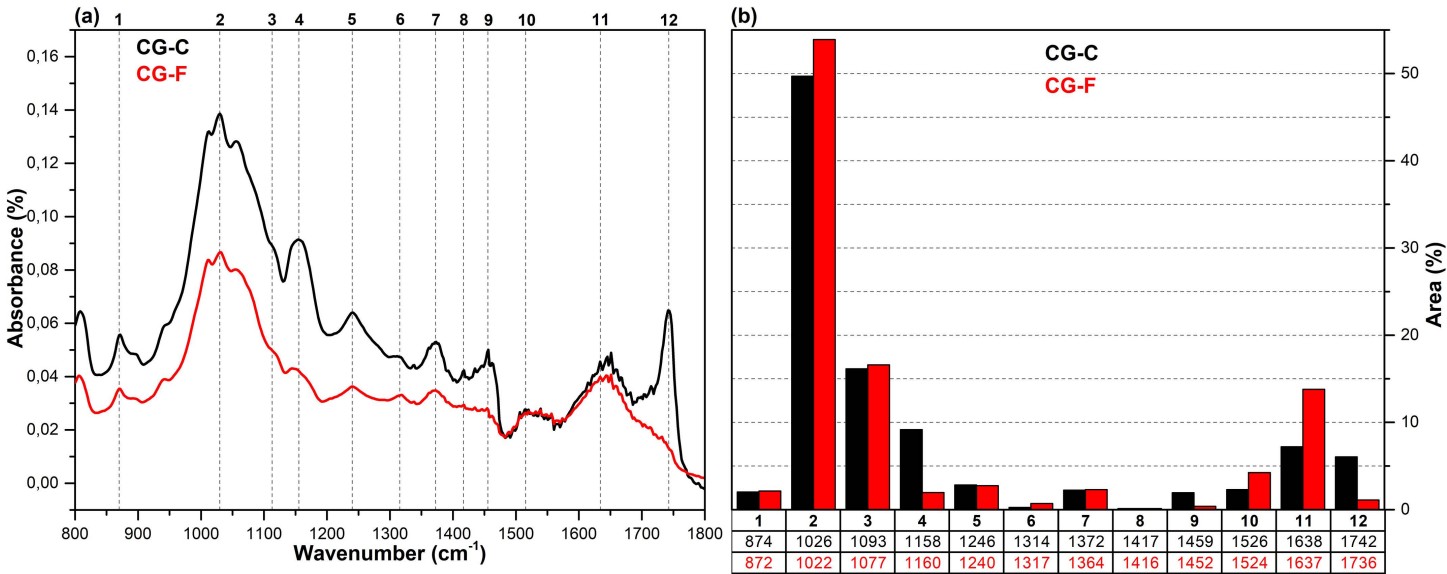

**Fig 8. FT-IR results. a)** FT-IR spectra of CG-C and CG-F; b) histograms of FT-IR peak areas processed using Gaussian curve fitting.

The bands between 1375–1360 cm$^{-1}$ and 1330–1310 cm$^{-1}$ correspond to symmetric and asymmetric C-H deformation vibrations of cellulose and hemicellulose, O-H deformation of kinic acid, $CH_2$ in-plane bending vibrations at the sixth carbon of crystalline cellulose, and C-H deformation of kinic acid, respectively [35,39]. After the fungal attack, the area of the first band (7) increased for the HH and CG samples, while it decreased for all other samples. Similarly, the area of the second band (6) increased for the CH, HH, and CG samples, but decreased in all others.

The bands between 1250–1230 cm$^{-1}$ correspond to C-O stretching vibrations of hemicellulose, lignin, and kinic acid (in CG samples) [39]. After the fungal attack, the area of this band (5) decreased for the all samples.

The weak shoulders observed between 1165–1155 cm$^{-1}$ and 1120–1075 cm$^{-1}$, along with the band observed between 1040–1020 cm$^{-1}$, are attributed to C–O–C vibrations of cellulose and hemicellulose (glycosidic linkages), quinic acid (in the CG sample), and C–OH bending vibrations [33,34,40]. After fungal attack, the area of the first band (4) decreased in the WS and CG samples, while it increased in the other samples. The significant decrease in the band area of the CG sample may be associated with the removal of quinic acids from the structure. The area of the second band (3) decreased in the HB, BW, RD, WS, and CH samples, while it increased in the OW, HH, and CG samples. The area of the third band (2) decreased in the HH sample, whereas it increased in all other samples.

The weak bands observed between 900 and 870 cm$^{-1}$ are attributed to $C_1$–O–$C_4$ β-(1–4) glycosidic bonds [40,41]. The area of these bands (1) increased in the CH, HH, and CG samples, while it decreased in all other samples.

*G. lucidum* was observed to exhibit degradative effects on lignocellulosic components, including cellulose, hemicellulose, lignin, and polysaccharides, as well as on chlorogenic acids present in spent coffee grounds. However, the extent of this degradation varied significantly depending on the substrate type. In particular, the degree of degradation in HH and CG substrates was considerably lower compared to the others, which likely hindered the biochemical environment required for fungal growth. Consequently, no mushroom development was observed on these substrates. In contrast, Akçay et al. [16] reported that *Pleurotus ostreatus*, unlike *G. lucidum*, was capable of degrading both HH and CG substrates, thereby enabling successful mushroom cultivation.

The FT-IR spectra (3600–800 cm$^{-1}$) of *G. lucidum* mushrooms cultivated on substrates of hazelnut branches (HB), beech wood (BW), oak wood (OW), rhododendron branches (RD), corn husk (CH), and wheat straw (WS), along with the

histograms of the peak areas obtained through Gaussian curve fitting applied to specific regions of these spectra, are presented in Fig 9.

The broad bands observed between 3500 and 3000 cm⁻¹ are attributed to O–H stretching vibrations due to intermolecular hydrogen bonds and humidity [42,43]. The bands observed at 2912 and 2854 cm⁻¹ are attributed to $CH_2$ asymmetric and symmetric stretching vibrations of lipids in *G. lucidum* mushrooms. These bands are associated with the structural features of triterpenoids, which are characteristic of *G. lucidum* mushrooms [42,44]. Similarly, the two bands observed at 1455 and 1380 cm⁻¹ are attributed to $CH_3$ in-plane and symmetric bending vibrations in the aliphatic structure of triterpene compounds. While these vibrations are primarily attributed to the structural properties of triterpenoids, it has also been suggested that protein and polysaccharide structures can influence these bands [43].

Puliga et al [31] reported that $CH_2$ bending vibrations in polysaccharides occur at 1455 cm⁻¹, while Wickramasinghe et al. [42] mentioned that there was the presence of β-glucan in the band at 1373 cm⁻¹. Analysis of the histograms corresponding to the bands (3, 4, 7) representing triterpene structures reveals that the peak area of the RD-M mushroom is greater than that of the other mushrooms.

The bands at 1635, 1535, and 1235 cm⁻¹ are attributed to amide I (C=O stretching), amide II (C–N, N–H stretching), and amide III (C–N, N–H stretching) in the protein structure of *G. lucidum* mushrooms, respectively [31,44,45]. Analysis of the histograms corresponding to these bands indicates that the RD-M mushroom exhibits the highest peak areas for the first (6) and second (5) bands, whereas the WS-M mushroom shows the largest peak area for the third band (2) compared to the other mushrooms.

The bands observed at 1150 and 1037 cm⁻¹ are attributed to C-O stretching vibrations of triterpene compounds and β-glucan of polysaccharides in the *G. lucidum,* as well as –OH and C–O–C stretching vibrations of sugar and polysaccharide structures (arabinan) [44]. Analysis of the histogram for the band at 1037 cm⁻¹ (1) reveals that the OW-M and CH-M mushrooms exhibit higher peak areas compared to the other mushrooms. The weak band observed at 889 cm⁻¹

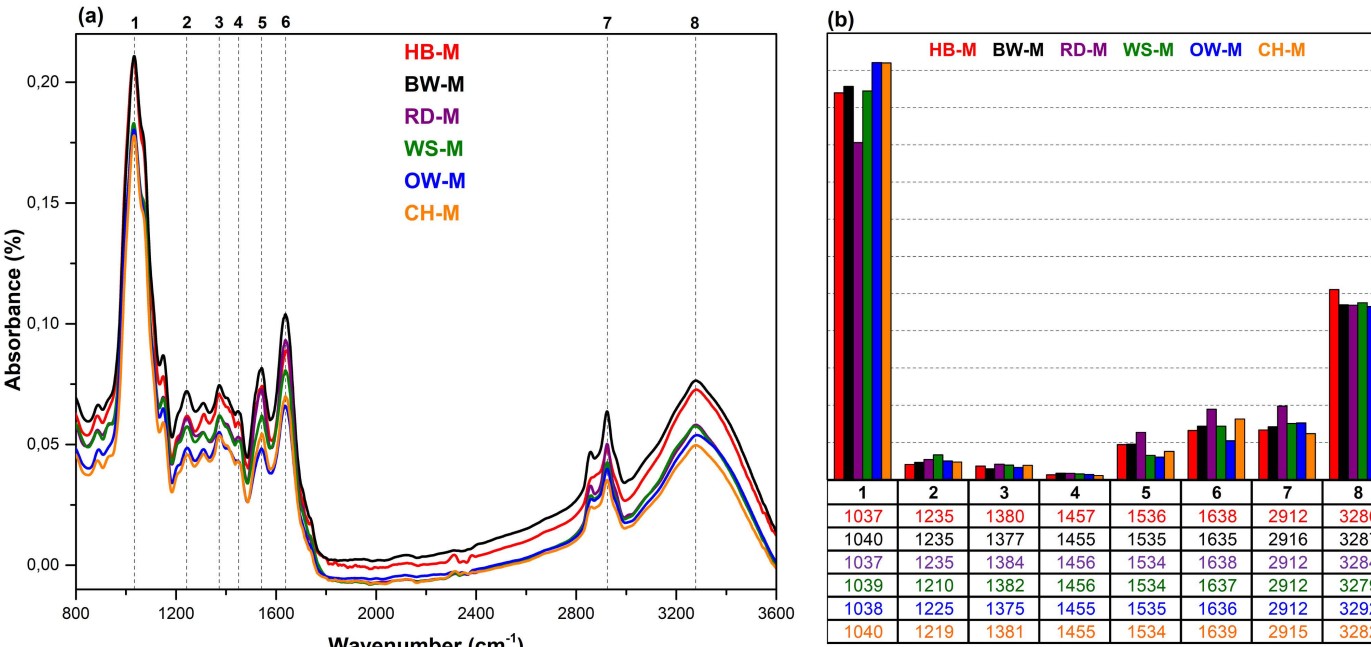

| 1 | 2 | 3 | 4 | 5 | 6 | 7 | 8 |
|---|---|---|---|---|---|---|---|
| 1037 | 1235 | 1380 | 1457 | 1536 | 1638 | 2912 | 3286 |
| 1040 | 1235 | 1377 | 1455 | 1535 | 1635 | 2916 | 3287 |
| 1037 | 1235 | 1384 | 1456 | 1534 | 1638 | 2912 | 3284 |
| 1039 | 1210 | 1382 | 1456 | 1534 | 1637 | 2912 | 3279 |
| 1038 | 1225 | 1375 | 1455 | 1535 | 1636 | 2912 | 3292 |
| 1040 | 1219 | 1381 | 1455 | 1534 | 1639 | 2915 | 3282 |

**Fig 9. FT-IR results. a)** FT-IR spectra of all mushroom samples; b) histograms of FT-IR peak areas processed using Gaussian curve fitting. HB-M: Mushroom fruit bodies cultivated from HB, other mushroom fruit bodies were coded similarly.

is correspons to the anomeric configuration of polysaccharides. This band is associated to the presence of 1,3 1,3-β-D-glucan as well as C–H deformation in polysaccharide structures [31].

When all *G. lucidum* mushrooms are examined collectively, it is evident that lipid, protein, and polysaccharide components are generally present in the mushroom structure, regardless of the substrate type (HB, BW, RD, WS, OW, CH). However, evaluation of the peak areas in the FT-IR spectra, calculated using Gaussian curve fitting, reveals quantitative differences in these components depending on the substrate. It can be inferred that mushrooms cultivated on the RD substrate contain higher levels of triterpenes and proteins compared to those grown on other substrates, whereas mushrooms grown on OW and CH substrates exhibit higher polysaccharide content. In this study, *G. lucidum* mushrooms could not be cultivated on HH and CG substrates, whereas Akçay et al [16] reported successful cultivation of *Pleurotus ostreatus* on both substrates.

### 3.4. Chemical analysis of the substrates before and after cultivation

Chemical analysis of the substrates before and after cultivation is shown in Table 5. When Table 5 was examined, it was found that holocellulose and pH values decreased while α–cellulose, extractives, and ash content values proportionately increased compared to previous values after cultivation. Significant differences were found in substrate contents. Among the holocellulose content of the substrates control, the highest amount was found in OW-C (80.71%), while the lowest was in HH-C (47.2%). Since OW contains high holocellulose, it can be said that OW-C gave a high mushroom yield. It can be said that the lack of mushroom fruit body formation in HH material is due to low cellulose content. The decrease in the amount of holocellulose after mushroom cultivation is due to the degradation activity of fungal enzymes of *G. lucidum*. The proportional increase in the amount of α-cellulose after fungal degradation (after mushroom cultivation) shows that it is due to the higher degradation of hemicellulose structures. Fungal enzymes act selectively in lignocellulosic structures, and that causes the structures to degrade at different rates. The proportional increases in the amount of extractive and ash show the proportional increase in the structure as a result of the degradation of organic structures and the inorganic structures not being degraded. Similar results have been determined in previous studies conducted by Zhang et al [46],

**Table 5. Chemical analysis of the substrates before and after cultivation (see S3 in S2 File for reference).**

| Substrates | Holocellulose (%) | α –Cellulose (%) | Extractives (%) | Ash (%) | pH |
|---|---|---|---|---|---|
| HB-C | 64.76±1.26[f] | 47.83±1.27[b] | 1.14±0.76[ab] | 2.54±0.27[d] | 7.01±0.11[h] |
| HB-F | 64.49±0.67[f] | 66.42±3.07[g] | 2.99±0.11[efgh] | 3.7±0.12[f] | 4.66±0.07[bc] |
| RD-C | 76.4±0.51[i] | 49.33±0.35[bc] | 3.56±0.54[fghi] | 1.61±0.22[b] | 4.68±0.13[bc] |
| RD-F | 67.06±0.43[g] | 54.58±3.86[de] | 3.64±0.52[ghi] | 2.12±0.15[c] | 4.35±0.06[a] |
| BW-C | 71.35±0.24[i] | 55.48±0.07[de] | 2.70±0.33[def] | 1.65±0.15[b] | 5.03±0.20[d] |
| BW-F | 66.6±0.24[g] | 58.9±0.7[ef] | 3±0.19[efgh] | 4.1±0.16[f] | 4.18±0.20[a] |
| OW-C | 80.71±0.35[j] | 53.68±0.32[cd] | 0.41±0.16[a] | 1.15±0.08[a] | 5.53±0.03[e] |
| OW-F | 69.89±0.76[h] | 55.05±0.85[de] | 1.44±0.59[bc] | 3±0.12[e] | 4.55±0.06[b] |
| HH-C | 47.2±0.32[a] | 49.36±0.42[bc] | 2.56±0.35[de] | 9.37±0.22[g] | 5.94±0.11[f] |
| HH-F | 49.6±0.36[b] | 58.96±8.58[ef] | 4.06±0.85[ii] | 10.5±0.05[h] | 5.41±0.09[e] |
| CG-C | 61.72±0.77[e] | 38.13±0.47[a] | 12.8±0.09[i] | 1.56±0.17[b] | 6.54±0.15[g] |
| CG-F | 51.85±0.40[c] | 37.8±0.39[a] | 2.74±0.19[def] | 2.96±0.21[e] | 5.07±0.06[d] |
| WS-C | 75.32±1.38[i] | 52.93±0.24[cd] | 1.99±0.66[cd] | 4.21±0.29[g] | 7.12±0.12[h] |
| WS-F | 66.76±0.54[g] | 60.48±0.94[f] | 4.12±0.38[i] | 8.55±0.05[f] | 4.76±0.14[c] |
| CH-C | 76.46±0.5[i] | 45.61±0.20[b] | 3.77±0.73[hi] | 3.1±0.66[e] | 5.18±0.10[d] |
| CH-F | 60.4±0.4[d] | 54.2±1.03[de] | 2.84±0.37[defg] | 5.88±0.12[h] | 5.13±0.07[d] |

Control samples (-C): Raw substrates before inoculation. Fungal degraded samples (-F): Spent substrates after *G. lucidum* harvest.

Akcay et al [47], and Akcay et al [16]. When extractive content was examined it was found that significant differences were detected between substrates ($p < 0.05$). Chemical analysis revealed significantly different extractive contents among substrates ($p < 0.05$), with spent coffee grounds (CG-C) showing the highest value (12.08%) and oak wood (OW-C) the lowest (0.41%). No significant difference was observed between OW-C and hazelnut branches (HB-C, 1.14%).

## 4. Conclusion

This study successfully cultivated *G. lucidum* using various agro-forestry residues, with oak wood (OW) demonstrating superior performance as the basal substrate (46 g/kg yield). Novel substrates like hazelnut branches (HB) and rhododendron (RD) demonstrated promising yields of 28.3–36.3 g/kg, indicating their potential as sustainable alternatives to conventional cultivation materials. The lignocellulosic materials used in this study could be promising alternative substrates for *G. lucidum* cultivation. However, hazelnut husk (HH) and spent coffee grounds (CG) failed to produce fruiting bodies, likely due to their unfavorable composition (low C/N ratios and high nitrogen content). Chemical analyses revealed significant substrate modifications post-cultivation, including holocellulose reduction (12–16%) and α-cellulose increase (25–39%), while FT-IR spectroscopy effectively tracked these structural changes. The mushrooms exhibited substrate-dependent variations in bioactive compounds, with beech wood-grown specimens showing the highest phenolic content (3.156 mg GAE/g). These findings advance sustainable mushroom cultivation practices while offering practical solutions for agricultural waste management. The study demonstrates that careful substrate selection can significantly affect both yield and product quality in medicinal mushroom production. The findings of this study suggest several promising directions for future research on *G. lucidum* cultivation. Optimizing substrate mixtures, particularly combinations like oak wood and hazelnut branches, could significantly enhance both yield and cost-efficiency while utilizing agricultural byproducts more effectively. A deeper investigation into the enzymatic mechanisms governing *G. lucidum*'s substrate preferences would provide valuable insights into its biodegradation capabilities, potentially enabling targeted improvements in cultivation practices. Finally, developing effective pretreatment methods for challenging waste materials such as hazelnut husks and spent coffee grounds could expand the range of viable substrates, further supporting sustainable mushroom production and waste valorization efforts.

### Field site access and permits

No specific permits were required for the collection of agricultural and forestry waste materials (hazelnut branches, wheat straw, *rhododendron* branches, etc.) used in this study, as these materials were obtained from:

✓ Privately owned farms in Düzce, Türkiye, with explicit verbal consent from the landowners for non-commercial research use.

✓ Local wood processing factories (for beech and oak sawdust), where materials were sourced as by-products.

✓ Publicly accessible areas (for *rhododendron* branches), where no protected species were involved

The study utilized agricultural/industrial by-products (e.g., pruning waste, husks, spent coffee grounds) classified as non-protected, non-endangered materials under Turkish environmental regulations. No fieldwork involved protected areas, endangered species, or ecological disturbances.

### Supporting information

**S1 File.** **S1.** Gaussian curve fitting analysis of HB-C. **S2.** Gaussian curve fitting analysis of HB-F. **S3.** Gaussian curve fitting analysis of BW-C. **S4.** Gaussian curve fitting analysis of BW-F. **S5.** Gaussian curve fitting analysis of OW-C. **S6.** Gaussian curve fitting analysis of OW-F. **S7.** Gaussian curve fitting analysis of RD-C. **S8.** Gaussian curve fitting analysis of RD-F. **S9.** Gaussian curve fitting analysis of WS-C. **S10.** Gaussian curve fitting analysis of WS-F. **S11.** Gaussian curve

fitting analysis of CH-C. **S12.** Gaussian curve fitting analysis of CH-F. **S13.** Gaussian curve fitting analysis of HH-C. **S14.** Gaussian curve fitting analysis of HH-F. **S15.** Gaussian curve fitting analysis of CG-C. **S16.** Gaussian curve fitting analysis of CG-F. **S17.** Gaussian curve fitting analysis of HB-M. **S18.** Gaussian curve fitting analysis of BW-M. **S19.** Gaussian curve fitting analysis of OW-M. **S20.** Gaussian curve fitting analysis of RD-M. **S21.** Gaussian curve fitting analysis of WS-M. **S22.** Gaussian curve fitting analysis of CH-M.
(RAR)

**S2 File.** **S1.** Details of substrate mixtures used for mushroom cultivation. **S2.** Total phenolic content of the substrates and mushrooms cultivated. **S3.** Chemical analysis of the substrates before and after cultivation.
(RAR)

## Author contributions

**Conceptualization:** Caglar Akcay, Recai Arslan, Faik Ceylan.

**Data curation:** Caglar Akcay, Recai Arslan, Faik Ceylan.

**Investigation:** Caglar Akcay, Recai Arslan, Faik Ceylan.

**Methodology:** Caglar Akcay, Recai Arslan, Faik Ceylan.

**Resources:** Caglar Akcay.

**Writing – original draft:** Caglar Akcay, Recai Arslan.

**Writing – review & editing:** Caglar Akcay.

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
