## [Decision Letter · Decision Letter 0]

PONE-D-25-12794Valorization of Various Lignocellulosic Wastes to Ganoderma lucidum (Reishi Mushroom) Cultivation and Their FT-IR AssessmentsPLOS ONE

Dear Dr. ARSLAN,

Thank you for submitting your manuscript to PLOS ONE. After careful consideration, we feel that it has merit but does not fully meet PLOS ONE’s publication criteria as it currently stands. Therefore, we invite you to submit a revised version of the manuscript that addresses the points raised during the review process.

We look forward to receiving your revised manuscript.

Kind regards,

Lee Seong

Academic Editor

PLOS ONE

“This research was financially supported by Directory of Scientific Research Projects of Düzce University (project number 2024.29.01.1446).”

Reviewers' comments:

Reviewer's Responses to Questions

**Comments to the Author**

1. Is the manuscript technically sound, and do the data support the conclusions?

Reviewer #1: Yes

Reviewer #2: Partly

2. Has the statistical analysis been performed appropriately and rigorously? 

Reviewer #1: Yes

Reviewer #2: Yes

3. Have the authors made all data underlying the findings in their manuscript fully available?

Reviewer #1: Yes

Reviewer #2: Yes

4. Is the manuscript presented in an intelligible fashion and written in standard English?

Reviewer #1: Yes

Reviewer #2: Yes

5. Review Comments to the Author

Reviewer #1: Comments to the Author:

Title: Valorization of Various Lignocellulosic Wastes to Ganoderma lucidum (Reishi Mushroom) Cultivation and Their FT-IR Assessments

Overview and general recommendation:

The manuscript deals with an important topic related to the valorization of various lignocellulosic wastes to Ganoderma lucidum (Reishi mushroom) cultivation and their FT-IR assessments. The manuscript technically sounds well and shows high novelty. However, it needs some linguistic adjustments. In this regard, the needed adjustments are highlighted in “Minor comments” section.

The Abstract section outlines clearly the problematic, aims, methodology and findings of the current study while reporting the main conclusions aroused. The Introduction section is well structured and aiming and underlines appropriately the whole subject under study. The aims of the study are also clear and understood. However, a sentence shall be added at the end of the Introduction section in which authors highlight how the findings of the present study are helpful. The Materials and Methods section is generally clear, well written, and encloses most of the information related to the adopted methodology, and statistical analysis. Only the adopted methods in paragraph 2.3. shall be briefly described. Although it shows a correct statistical representation, the Results and discussion section needs adjustments. The scientific analysis of the findings should be well improved. Percentages of variation (improvements/decreases) should be highlighted. However, the authors discussed appropriately the findings of their study and compared them with previously published ones in literature. A more concise Conclusion section shall be provided in which authors summarize the findings of their study and suggest further related research being based on the raised assumptions.

My comments and queries for authors are detailed below in “Major comments” and “Minor comments” sections.

1.1. Major comments:

1- The manuscript needs some linguistic adjustments. Most needed adjustments are highlighted in “Minor comments” section.

2- 1. Introduction: A sentence shall be added at the end of the Introduction section in which authors highlight how the findings of the present study are helpful.

3- 2. Materials and Methods, 2.3. Chemical analysis of substrates: Kindly describe briefly the adopted methods.

4- 3. Results and discussion: The scientific analysis of the findings should be well improved. Percentages of variation (improvements/decreases) should be highlighted.

5- 4. Conclusion: A more concise Conclusion section shall be provided in which authors summarize the findings of their study and suggest further related research being based on the raised assumptions.

1.2. Minor comments:

6- Abstract: Page 1, line 16: Kindly adjust as follow: “explored”.

7- Abstract: Page 1, lines 18–19: “G. lucidum… viability”: Kindly move this sentence to the beginning of the Abstract section.

8- Abstract: Page 2, line 24: Kindly replace “Composts” by “Substrates”.

9- 1. Introduction: Page 3, lines 46 and 49: Reference 3 is relatively old; accordingly, kindly replace it by a more recent one (last five years of publication).

10- 1. Introduction: Page 3, line 54: Reference 5 is relatively old; accordingly, kindly replace it by a more recent one (last five years of publication).

11- 1. Introduction: Page 3, lines 65–66: “In many… [9]”: The reference used for this statement is relatively old; accordingly, kindly replace it by the following recent and reliable one: “doi:10.30486/IJROWA.2023.1964536.1513”.

12- 2. Materials and Methods, 2.1. Preparation of mixtures from waste materials: Page 5, line 92: Kindly adjust the title of this paragraph as follow: “2.1. Preparation of mixtures from waste materials”.

13- 2. Materials and Methods, 2.1. Preparation of mixtures from waste materials: Page 5, line 110: Kindly replace “composts” by “substrates”.

14- 2. Materials and Methods, 2.2. Incubation and mushroom harvesting: Page 6, line 124: Kindly replace “compost” by “substrate”.

15- 2. Materials and Methods, 2.2. Incubation and mushroom harvesting: Page 6, Table 1: Did you perform previously a pilot study in order to take the choice of the provided mixtures substrates’ proportions?? You shall mention the reason behind the choice of proportions.

16- 2. Materials and Methods, 2.5. Elemental analysis: Page 8, line 147: Kindly remove “(2024)”.

17- 2. Materials and Methods, 2.7. Statistical analysis: Page 8, line 159: Kindly adjust the title of this paragraph as follow: “2.7. Statistical analysis”.

18- 3. Results and discussion, 3.1. Spawn run time, earliness, yield, biological efficiency and dry matter: Page 9, lines 168–169: “Spawn… Table 2”: The sentence is badly written in standard English; accordingly, kindly reformulate it.

19- 3. Results and discussion, 3.1. Spawn run time, earliness, yield, biological efficiency and dry matter: Page 9, lines 176 and 182: Kindly adjust as follow: “by Atila et al. [2]”.

20- 3. Results and discussion, 3.1. Spawn run time, earliness, yield, biological efficiency and dry matter: Page 9, lines 182–183: Kindly adjust as follow: “by Gurung et al. [23] and Roy et al. [24]”.

21- 3. Results and discussion, 3.1. Spawn run time, earliness, yield, biological efficiency and dry matter: Page 10, line 191: Kindly adjust as follow: “prepared with”.

22- 3. Results and discussion, 3.1. Spawn run time, earliness, yield, biological efficiency and dry matter: Page 10, lines 195–196: Kindly adjust as follow: “by Akcay et al. [16]”.

23- 3. Results and discussion, 3.1. Spawn run time, earliness, yield, biological efficiency and dry matter: Page 10, line 200: Kindly remove “Carrasco-Cabrera, Bell”.

24- 3. Results and discussion, 3.1. Spawn run time, earliness, yield, biological efficiency and dry matter: Page 10, line 208: Kindly replace “similar to” by “comparable with”.

25- 3. Results and discussion, 3.2. Total organic content (TOC), phenolic content (TPC) and elemental composition: Page 12, line 232: Kindly mention the lowest value percentage here.

26- 3. Results and discussion, 3.2. Total organic content (TOC), phenolic content (TPC) and elemental composition: Page 12, line 233: Kindly replace “similar” by “comparable”.

27- 3. Results and discussion, 3.2. Total organic content (TOC), phenolic content (TPC) and elemental composition: Page 13, line 247: Kindly adjust as follow: “cultivated on”.

28- 3. Results and discussion, 3.2. Total organic content (TOC), phenolic content (TPC) and elemental composition: Page 13, lines 248–249: Kindly adjust the sentence as follow: “Modi et al. [29] reported that…”

29- 3. Results and discussion, 3.2. Total organic content (TOC), phenolic content (TPC) and elemental composition: Page 13, lines 249–250: Kindly adjust the sentence as follow: “Atila et al. [2] found…”

30- 3. Results and discussion, 3.2. Total organic content (TOC), phenolic content (TPC) and elemental composition: Page 13, lines 251–252: Kindly adjust the sentence as follow: “Demirci et al. [30] reported that…”

31- 3. Results and discussion, 3.2. Total organic content (TOC), phenolic content (TPC) and elemental composition: Page 13, lines 254–255: Kindly adjust as follow: “in this study” and “reported by Modi et al. [29] and Atila et al. [2]”.

32- 3. Results and discussion, 3.2. Total organic content (TOC), phenolic content (TPC) and elemental composition: Page 13, line 255: Kindly replace “similar” by “comparable” and adjust as follow: “reported by Demirci et al. [30]”.

33- 3. Results and discussion, 3.3. FT-IR spectroscopy and biodegradation properties: Page 17, line 320: Kindly adjust the sentence as follow: “In contrast, Akcay et al. [16] reported that…”

34- 3. Results and discussion, 3.3. FT-IR spectroscopy and biodegradation properties: Page 18, lines 346–347: Kindly adjust as follow: “by Akcay et al. [16]”.

35- 3. Results and discussion, 3.3. FT-IR spectroscopy and biodegradation properties: Page 18, line 352: Kindly replace “are” by “were”.

36- 3. Results and discussion, 3.3. FT-IR spectroscopy and biodegradation properties: Page 20, lines 379–380: Kindly adjust the sentence as follow: “Puliga et al. [31] reported that… while Wickramasinghe et al. [42] mentioned that…”

37- 3. Results and discussion, 3.3. FT-IR spectroscopy and biodegradation properties: Page 20, line 387: Kindly adjust as follow: “corresponds to”.

38- 3. Results and discussion, 3.3. FT-IR spectroscopy and biodegradation properties: Page 21, lines 394–395: Kindly adjust the sentence as follow: “… Akcay et al. [16] observed that…”

39- 3. Results and discussion, 3.4. Chemical analysis of the substrates before and after cultivation: Page 21, line 401: Kindly adjust as follow: “compared to”.

40- 3. Results and discussion, 3.4. Chemical analysis of the substrates before and after cultivation: Page 22, line 414: Kindly adjust the sentence as follow: “… conducted by Zhang et al. [46], Akcay et al. [47], and Akcay et al. [16]”.

41- 3. Results and discussion, 3.4. Chemical analysis of the substrates before and after cultivation: Page 22, line 415: Kindly adjust as follow: “were detected”.

42- 3. Results and discussion, 3.4. Chemical analysis of the substrates before and after cultivation: Page 22, line 416–417: “The highest… (1.14%)”: The sentence is badly written in standard English; accordingly, kindly reformulate it.

Reviewer #2: The presented topic dealt with valorization of diverse lignocellulosic biomass to cultivate Ganoderma lucidum (Reishi Mushroom). The topic worth investigation. However, the Ms has some flaws with regards to

1) Methodology and the design of the experiment (see text): How composting was done? For how long the composting process done? How C: N ratio was maintained? This step is fundamentally crucial and needs explanation! How was CO2 level maintained? For how long the inoculated substrate incubated at 24oC? How the intensity of light was monitored? Was there casing material? If yes, what medium was used as casing material? Why the proportion was kept constant at 91:9/75:25? Why not 70:30/65:35/45:55 or 25:75 or other proportions????? Actually, many possible combinations could have been done!! This is a serious limitation/gap in this work!

2) Results and Discussion

No procedures in the methodology but findings reported in this section (see text). Almost no Discussion for presented data/findings.

3) The author/s/ benefit from the technical language improvement

6. PLOS authors have the option to publish the peer review history of their article (what does this mean? ). If published, this will include your full peer review and any attached files.

**Do you want your identity to be public for this peer review?** For information about this choice, including consent withdrawal, please see our Privacy Policy .

Reviewer #1: No

Reviewer #2: **Yes: ** Diriba Muleta

---

## [Author Response · Author response to Decision Letter 1]

18 Apr 2025

Valorization of Various Lignocellulosic Wastes to Ganoderma lucidum (Reishi Mushroom) Cultivation and Their FT-IR Assessments

The authors have carefully addressed all reviewer comments in the revised manuscript with changes clearly marked in red and with Tracked Changes for text revisions, and blue for responses to reviewer queries in this document.

REVIEWER 1

Comments to the Author:Title: Valorization of Various Lignocellulosic Wastes to Ganoderma lucidum (Reishi Mushroom) Cultivation and Their FT-IR Assessments

Overview and general recommendation: The manuscript deals with an important topic related to the valorization of various lignocellulosic wastes to Ganoderma lucidum (Reishi mushroom) cultivation and their FT-IR assessments. The manuscript technically sounds well and shows high novelty. However, it needs some linguistic adjustments. In this regard, the needed adjustments are highlighted in “Minor comments” section.

The Abstract section outlines clearly the problematic, aims, methodology and findings of the current study while reporting the main conclusions aroused.

The Introduction section is well structured and aiming and underlines appropriately the whole subject under study. The aims of the study are also clear and understood. However, a sentence shall be added at the end of the Introduction section in which authors highlight how the findings of the present study are helpful.

The Materials and Methods section is generally clear, well written, and encloses most of the information related to the adopted methodology, and statistical analysis. Only the adopted methods in paragraph 2.3. shall be briefly described.

Although it shows a correct statistical representation, the Results and discussion section needs adjustments. The scientific analysis of the findings should be well improved. Percentages of variation (improvements/decreases) should be highlighted. However, the authors discussed appropriately the findings of their study and compared them with previously published ones in literature.

A more concise Conclusion section shall be provided in which authors summarize the findings of their study and suggest further related research being based on the raised assumptions.

My comments and queries for authors are detailed below in “Major comments” and “Minor comments” sections.1.1. Major comments:1- The manuscript needs some linguistic adjustments. Most needed adjustments are highlighted in “Minor comments” section.2- 1.

Introduction: A sentence shall be added at the end of the Introduction section in which authors highlight how the findings of the present study are helpful.

Following sentence was added to the introduction section

The findings of this study provide valuable insights into the efficient utilization of agricultural and forest wastes for G. lucidum cultivation, offering a sustainable solution for waste management and the production of high-value medicinal mushrooms.

3- 2. Materials and Methods, 2.3. Chemical analysis of substrates: Kindly describe briefly the adopted methods.

Methods were described as follow

Extractives: Determined according to TAPPI standards (T 204 cm-07) using solvent extraction (a toluene: acetone: ethanol mixture [4/1/1, (v/v)] for 6 h using a soxhlet extractor) to remove non-structural components like resins, fats, and waxes (Akçay and Yalçın 2021).

Holocellulose: Measured via the chlorite method (Wise et al., 1952), which selectively removes lignin while retaining cellulose and hemicellulose.

α-Cellulose: Isolated by treating holocellulose with NaOH (17.5% and 8.3%) to dissolve hemicellulose, following TAPPI T203 cm-09 (TAPPI, 2009).

Ash Content: Calculated by combusting substrates at 575°C until constant weight (TAPPI T211 om-16) to determine inorganic residues (Kacar, 1994).

pH: Measured in a 1:10 (w/v) substrate-water suspension using a calibrated pH meter (Kacar, 1994).

4- 3. Results and discussion: The scientific analysis of the findings should be well improved. Percentages of variation (improvements/decreases) should be highlighted.

following sentences were added to the text

In Ganoderma lucidum cultivation, oak wood (OW) is widely recognized as the traditional basal substrate due to its optimal lignin content and C/N ratio (~50) for fruiting body formation. Compared to alternative substrates like wheat straw (WS), OW demonstrated superior performance in this study, yielding 46 g/kg—a 148% increase over WS (18.5 g/kg).

TPC

(~87% greater than those from WS).

5- 4. Conclusion: A more concise Conclusion section shall be provided in which authors summarize the findings of their study and suggest further related research being based on the raised assumptions.1.2. Minor comments:

Both reviewers are right, conclusion section was re-written according to their recommendations

This study successfully cultivated G. lucidum using various Agro-forestry residues, with oak wood (OW) demonstrating superior performance as the basal substrate (46 g/kg yield). Novel substrates like hazelnut branches (HB) and rhododendron (RD) demonstrated promising yields of 28.3-36.3 g/kg, indicating their potential as sustainable alternatives to conventional cultivation materials. However, hazelnut husk (HH) and spent coffee grounds (CG) failed to produce fruiting bodies, likely due to their unfavorable composition (low C/N ratios and high nitrogen content). Chemical analyses revealed significant substrate modifications post-cultivation, including holocellulose reduction (12-16%) and α-cellulose increase (25-39%), while FT-IR spectroscopy effectively tracked these structural changes. The mushrooms exhibited substrate-dependent variations in bioactive compounds, with beech wood-grown specimens showing the highest phenolic content (3.156 mg GAE/g). These findings advance sustainable mushroom cultivation practices while offering practical solutions for agricultural waste management. The study demonstrates that careful substrate selection can significantly affect both yield and product quality in medicinal mushroom production. Based on these findings, the following research directions are recommended;

1. Optimizing substrate mixtures (e.g., OW+HB blends) to enhance yield and cost-efficiency,

2. Investigating the enzymatic mechanisms behind G. lucidum's substrate preferences, and

3. Developing pretreatment methods for challenging wastes like HH and CG.

6- Abstract: Page 1, line 16: Kindly adjust as follow: “explored”.7- Abstract: Page 1, lines 18–19: “G. lucidum… viability”: Kindly move this sentence to the beginning of the Abstract section.8- Abstract: Page 2, line 24: Kindly replace “Composts” by “Substrates”.9- 1.

Abstract revisions were done on the manuscript.

Introduction: Page 3, lines 46 and 49: Reference 3 is relatively old; accordingly, kindly replace it by a more recent one (last five years of publication).

Anusiya G, Gowthama Prabu U, Yamini NV, Sivarajasekar N, Rambabu K, Bharath G, Banat F. A review of the therapeutic and biological effects of edible and wild mushrooms. Bioengineered. 2021;12(2):11239-11268. doi: 10.1080/21655979.2021.2001183

10- 1. Introduction: Page 3, line 54: Reference 5 is relatively old; accordingly, kindly replace it by a more recent one (last five years of publication).

Kumar, A. Ganoderma Lucidum: A traditional chinese medicine used for curing tumors. Int J Pharm Pharm Sci. 2021; 13(3), 1-13.

11- 1. Introduction: Page 3, lines 65–66: “In many… [9]”: The reference used for this statement is relatively old; accordingly, kindly replace it by the following recent and reliable one: “doi:10.30486/IJROWA.2023.1964536.1513”.

Abou Fayssal, S., & Yordanova, M. H. (2024). Effect of substrate temperature and stages duration on recycling of agro-industrial residues through Pleurotus ostreatus production. International Journal of Recycling of Organic Waste in Agriculture, 12(4). https://doi.org/10.30486/ijrowa.2023.1964536.1513

12- 2. Materials and Methods, 2.1. Preparation of mixtures from waste materials: Page 5, line 92: Kindly adjust the title of this paragraph as follow: “2.1. Preparation of mixtures from waste materials”.13- 2. Materials and Methods, 2.1. Preparation of mixtures from waste materials: Page 5, line 110: Kindly replace “composts” by “substrates”.14- 2. Materials and Methods, 2.2. Incubation and mushroom harvesting: Page 6, line 124: Kindly replace “compost” by “substrate”.

Done

15- 2. Materials and Methods, 2.2. Incubation and mushroom harvesting: Page 6, Table 1: Did you perform previously a pilot study in order to take the choice of the provided mixtures substrates’ proportions?? You shall mention the reason behind the choice of proportions.1

Following passage was added to the text

The substrate proportions were carefully selected based on established cultivation protocols and waste valorization objectives. Control groups utilized a standardized 91% substrate + 9% wheat bran [10], where wheat bran provided optimal nitrogen supplementation. For hazelnut branch (HB) mixtures, a 75% HB + 25% other materials ratio was implemented to specifically address two key considerations: the material's significance as Turkey's prominent agricultural waste (1.7 million tons/year; [16]) with favorable lignocellulosic composition for fungal growth, and (2) the need to evaluate potential synergistic effects of substrate combinations while preserving HB's structural integrity. This experimental design enabled systematic comparison between conventional substrates and novel waste-based alternatives.

6- 2. Materials and Methods, 2.5. Elemental analysis: Page 8, line 147: Kindly remove “(2024)”.17- 2. Materials and Methods, 2.7. Statistical analysis: Page 8, line 159: Kindly adjust the title of this paragraph as follow: “2.7. Statistical analysis”.

Done

18- 3. Results and discussion, 3.1. Spawn run time, earliness, yield, biological efficiency and dry matter: Page 9, lines 168–169: “Spawn… Table 2”: The sentence is badly written in standard English; accordingly, kindly reformulate it.

It was re written as fallow.

Table 2 presents the spawn run time, earliness (days to first harvest), yield, biological efficiency (BE), and dry matter content of mushrooms cultivated on different substrates.

19- 3. Results and discussion, 3.1. Spawn run time, earliness, yield, biological efficiency and dry matter: Page 9, lines 176 and 182: Kindly adjust as follow: “by Atila et al. [2]”.20- 3. Results and discussion, 3.1. Spawn run time, earliness, yield, biological efficiency and dry matter: Page 9, lines 182–183: Kindly adjust as follow: “by Gurung et al. [23] and Roy et al. [24]”.21- 3. Results and discussion, 3.1. Spawn run time, earliness, yield, biological efficiency and dry matter: Page 10, line 191: Kindly adjust as follow: “prepared with”.22- 3. Results and discussion, 3.1. Spawn run time, earliness, yield, biological efficiency and dry matter: Page 10, lines 195–196: Kindly adjust as follow: “by Akcay et al. [16]”.23- 3. Results and discussion, 3.1. Spawn run time, earliness, yield, biological efficiency and dry matter: Page 10, line 200: Kindly remove “Carrasco-Cabrera, Bell”.24- 3. Results and discussion, 3.1. Spawn run time, earliness, yield, biological efficiency and dry matter: Page 10, line 208: Kindly replace “similar to” by “comparable with”.25- 3. Results and discussion, 3.2. Total organic content (TOC), phenolic content (TPC) and elemental composition: Page 12, line 232: Kindly mention the lowest value percentage here.26- 3. Results and discussion, 3.2. Total organic content (TOC), phenolic content (TPC) and elemental composition: Page 12, line 233: Kindly replace “similar” by “comparable”.27- 3. Results and discussion, 3.2. Total organic content (TOC), phenolic content (TPC) and elemental composition: Page 13, line 247: Kindly adjust as follow: “cultivated on”.28- 3. Results and discussion, 3.2. Total organic content (TOC), phenolic content (TPC) and elemental composition: Page 13, lines 248–249: Kindly adjust the sentence as follow: “Modi et al. [29] reported that…”29- 3. Results and discussion, 3.2. Total organic content (TOC), phenolic content (TPC) and elemental composition: Page 13, lines 249–250: Kindly adjust the sentence as follow: “Atila et al. [2] found…”30- 3. Results and discussion, 3.2. Total organic content (TOC), phenolic content (TPC) and elemental composition: Page 13, lines 251–252: Kindly adjust the sentence as follow: “Demirci et al. [30] reported that…”31- 3. Results and discussion, 3.2. Total organic content (TOC), phenolic content (TPC) and elemental composition: Page 13, lines 254–255: Kindly adjust as follow: “in this study” and “reported by Modi et al. [29] and Atila et al. [2]”.32- 3. Results and discussion, 3.2. Total organic content (TOC), phenolic content (TPC) and elemental composition: Page 13, line 255: Kindly replace “similar” by “comparable” and adjust as follow: “reported by Demirci et al. [30]”.33- 3. Results and discussion, 3.3. FT-IR spectroscopy and biodegradation properties: Page 17, line 320: Kindly adjust the sentence as follow: “In contrast, Akcay et al. [16] reported that…”34- 3. Results and discussion, 3.3. FT-IR spectroscopy and biodegradation properties: Page 18, lines 346–347: Kindly adjust as follow: “by Akcay et al. [16]”.35- 3. Results and discussion, 3.3. FT-IR spectroscopy and biodegradation properties: Page 18, line 352: Kindly replace “are” by “were”.36- 3. Results and discussion, 3.3. FT-IR spectroscopy and biodegradation properties: Page 20, lines 379–380: Kindly adjust the sentence as follow: “Puliga et al. [31] reported that… while Wickramasinghe et al. [42] mentioned that…”37- 3. Results and discussion, 3.3. FT-IR spectroscopy and biodegradation properties: Page 20, line 387: Kindly adjust as follow: “corresponds to”.38- 3. Results and discussion, 3.3. FT-IR spectroscopy and biodegradation properties: Page 21, lines 394–395: Kindly adjust the sentence as follow: “… Akcay et al. [16] observed that…”39- 3. Results and discussion, 3.4. Chemical analysis of the substrates before and after cultivation: Page 21, line 401: Kindly adjust as follow: “compared to”.40- 3. Results and discussion, 3.4. Chemical analysis of the substrates before and after cultivation: Page 22, line 414: Kindly adjust the sentence as follow: “… conducted by Zhang et al. [46], Akcay et al. [47], and Akcay et al. [16]”.41- 3. Results and discussion, 3.4. Chemical analysis of the substrates before and after cultivation: Page 22, line 415: Kindly adjust as follow: “were detected”.

All revisions were done in the manuscript

42- 3. Results and discussion, 3.4. Chemical analysis of the substrates before and after cultivation: Page 22, line 416–417: “The highest… (1.14%)”: The sentence is badly written in standard English; accordingly, kindly reformulate it.

Following sentences were added.

Chemical analysis revealed significantly different extractive contents among substrates (p < 0.05), with spent coffee grounds (CG-C) showing the highest value (12.08%) and oak wood (OW-C) the lowest (0.41%). No significant difference was observed between OW-C and hazelnut branches (HB-C, 1.14%)

The authors sincerely thank the reviewer 1 for the exceptionally thorough and constructive evaluation of our manuscript. Your careful attention to detail and valuable suggestions have significantly improved the quality of this work. We are truly grateful for the time and expertise you have dedicated to reviewing our paper.

REVIEWER 2:

The presented topic dealt with valorization of diverse lignocellulosic biomass to cultivate Ganoderma lucidum (Reishi Mushroom). The topic worth investigation. However, the Ms has some flaws with regards to1) Methodology and the design of the experiment (see text):

Thank you for the finding study valuable.

How composting was done? For how long the composting process done? How C: N ratio was maintained? This step is fundamentally crucial and needs explanation! How was CO2 level maintained?

We sincerely apologize for any confusion regarding terminology in our original manuscript. Upon careful reconsideration, we have replaced all instances of "compost" with "substrate" throughout the text to accurately reflect our methodology. The mixtures used in this study were:

1. Not traditionally composted (no extended microbial decomposit

---

## [Decision Letter · Decision Letter 1]

PONE-D-25-12794R1Valorization of Various Lignocellulosic Wastes to Ganoderma lucidum (Reishi Mushroom) Cultivation and Their FT-IR Assessments

PLOS ONE

Dear Dr. ARSLAN,

Thank you for submitting your manuscript to PLOS ONE. After careful consideration, we feel that it has merit but does not fully meet PLOS ONE’s publication criteria as it currently stands. Therefore, we invite you to submit a revised version of the manuscript that addresses the points raised during the review process.

=====

Comment from the Editorial Office: Reviewer 2 informed us that they submitted the wrong review for your manuscript. Please disregard their comments and focus your revisions on the comments of Reviewers 1, 3, and 4. Please contact plosone@plos.org if you have any questions or concerns. Thank you.

=====

We look forward to receiving your revised manuscript.

Kind regards,

Lee Seong

Academic Editor

PLOS ONE

Reviewers' comments:

Reviewer's Responses to Questions

**Comments to the Author**

1. If the authors have adequately addressed your comments raised in a previous round of review and you feel that this manuscript is now acceptable for publication, you may indicate that here to bypass the “Comments to the Author” section, enter your conflict of interest statement in the “Confidential to Editor” section, and submit your "Accept" recommendation.

Reviewer #1: All comments have been addressed

Reviewer #2: (No Response)

Reviewer #3: (No Response)

Reviewer #4: (No Response)

2. Is the manuscript technically sound, and do the data support the conclusions?

Reviewer #1: Yes

Reviewer #2: Partly

Reviewer #3: Yes

Reviewer #4: (No Response)

3. Has the statistical analysis been performed appropriately and rigorously? 

Reviewer #1: Yes

Reviewer #2: No

Reviewer #3: No

Reviewer #4: (No Response)

4. Have the authors made all data underlying the findings in their manuscript fully available?

Reviewer #1: Yes

Reviewer #2: Yes

Reviewer #3: Yes

Reviewer #4: Yes

5. Is the manuscript presented in an intelligible fashion and written in standard English?

Reviewer #1: Yes

Reviewer #2: No

Reviewer #3: No

Reviewer #4: No

6. Review Comments to the Author

Reviewer #1: Comments to the Author:

Title: Valorization of Various Lignocellulosic Wastes to Ganoderma lucidum (Reishi Mushroom) Cultivation and Their FT-IR Assessments

Overview and general recommendation:

Authors have made all needed improvements to their manuscript and are well thanked for that. Therefore, based on the overall evaluation of the manuscript, I find it well suitable for publication in current form.

Reviewer #2: The author/s/ tried to isolate a novel and potent species of actinomycetes against common human bacterial pathogens. However, the work did not consider a large sample size that may lead to isolation of potent and novel Streptomyces species. The obtained result is also not encouraging since the selected isolated showed a moderate inhibition against the test pathogens. The study also did not include fungal pathogens. For details see the text.

Reviewer #3: In this work by Caglar Akcay and colleagues, the cultivation of the medicinal mushroom Ganoderma lucidum was tested on various agricultural residues. The degradative capacity of G. lucidum on different substrates was assessed using Fourier-transform infrared (FTIR) spectroscopy. Furthermore, the chemical composition of both the substrates and the fruiting bodies obtained after cultivation was investigated.

Overall, the manuscript requires substantial improvements, particularly in methodology, language and English style. Specifically:

Abstract – Revisions are necessary to include results concerning the chemical composition of both the substrates and the harvested fruiting bodies.

Line 16: The full scientific name Ganoderma lucidum along with the authority should be provided at its first mention, both in the abstract and the main text. The common name "Reishi" should be inserted in parentheses immediately afterward and removed from line 18.

Introduction –

Line 40: The full name and authority for Ganoderma lucidum should be provided at first mention. Additionally, when the name appears at the beginning of a sentence (see line 56), it should be written out in full.

Line 77: Replace “Turkiye” with “Turkey.”

Lines 81–82: Insert the authority for Pleurotus ostreatus.

Lines 82–84: While the authors provided context for the selection of hazelnut branches as a substrate, no justification is given for the choice of rhododendron. Context should be added to support this choice.

Materials and Methods –

Line 97: Please insert the authority for Fagus orientalis.

Lines 98–102: Move the substrate abbreviations to lines 96–98, where they are first mentioned.

Line 102: Replace “Turkiye” with “Turkey.”

Lines 110–116: This section should be moved to the Introduction, as it explains the objectives of the study.

Line 127: Replace “Turkiye” with “Turkey.”

Lines 127–128: Replace "compost" with "substrate."

Lines 138–139: After colonization, were the bags opened to stimulate primordia formation, or were they kept closed until primordia emerged and then opened to promote primordia development and fruiting body expansion? Please clarify.

Lines 145–150: This information should be rewritten in a more discursive manner rather than presented as bullet point.

Line 154: What was the dry weight of the different formulated substrates? This is essential for calculating biological efficiency. Please provide this information.

Lines 156–166: This paragraph should be written in a more discursive manner and should include more detailed information about the methods used.

Lines 192–196: Please provide additional details on how the obtained spectra were processed. How many replicates per treatment were conducted? The figures (Figures 1–9) suggest that baseline correction was not performed. To better highlight significant changes in the substrates before and after fungal growth, it is recommended that the authors include histograms of peak areas (%) processed using curve fitting in the region from 1800 to 1200 cm⁻¹.

Lines 203–216: Please remove this paragraph, as it does not belong in the Materials and Methods section. At most, it may be included at the end of the manuscript.

Results and Discussion – This section needs further development. The authors should discuss their findings in greater depth and include more references.

Line 227: “Some studies…” – Which studies? Please provide references.

Lines 233–236: What explanation do the authors provide for this earlier onset of fruiting?

Lines 245–247: Beech sawdust is also widely used as a cultivation substrate.

Lines 261–262: This sentence should be moved to the Conclusions section.

Lines 290–291: What is the optimal C/N ratio for the growth and fruiting of G. lucidum?

Lines 292–293: On what basis do the authors make this claim? Supporting literature and

discussion are needed.

Lines 305–311: The authors state that TPC values in mushrooms may vary depending on the solvent used for extraction. However, given the same methodology and solvent, how do they explain the significantly lower values reported here compared to those in the literature?

Line 323: Insert the authority for Lentinus edodes.

Table 4: The footnote is unclear and should be revised for clarity.

Conclusions –

Lines 492–495: This text should be rewritten in a more discursive manner.

Reviewer #4: Reviewer’s comments on PONE-D-25-12794R1

Title: Suitable

Abstract: Alright

Introduction: Line 58: Make it forest residue and not forestry residue. Line 63: Please unbundle what you mean by energy or food. What do authors mean by biodegradation properties of G. lucidum? If the novelty statement of the manuscript suggests cultivation of the medicinal mushroom on lignocellulosic waste, authors should highlight in the last but one paragraph of the ‘Introduction’ section what substrates have been in use before now and why they think lignocellulosic substrates should replace them.

Materials and methods: Lines 103-105: Why the variations in substrate particle sizes? Lines 106-108: Please state what informed the substrate mixtures and/or cite an appropriate reference for it. What do authors mean by ‘standardized 91% substrate and 9% wheat bran? As far as I know, wheat bran is also a substrate which contains 46% of non-starch polysaccharides, including arabinoxylan (70%), cellulose (24%) and beta-glucan (6%), and it also contains minor amounts of glucoglucomannan and arabinogalactan (Carre and Brillouet, 1986). What is with wheat bran serving as a nitrogen source (supplementation)? Even if it can, it should not serve as a primary nitrogen source. Line 118: To what level was the pH adjusted? What was the 90 minutes autoclaving for? How did authors arrive at that? Please remove all horizontal and vertical lines from Table 1 except the top two horizontal lines for headers and bottom horizontal line for closure. Line 129: please make the phrase ‘mixed homogeneously’ read ‘mixed to homogeneity’. Lines 132-133: Cite reference. Line 136: How did authors raise the incubation to a range between 28 and 30C? Please, pick a temperature value to which the preparation was incubated and why. Cite a reference.

Results: Please remove all horizontal and vertical lines from all tables as earlier recommended. What is the relationship between alpha-cellulose and hemicellulose?

7. PLOS authors have the option to publish the peer review history of their article (what does this mean? ). If published, this will include your full peer review and any attached files.

**Do you want your identity to be public for this peer review?** For information about this choice, including consent withdrawal, please see our Privacy Policy .

Reviewer #1: No

Reviewer #2: **Yes: ** Diriba Muleta

Reviewer #3: No

Reviewer #4: No

---

## [Author Response · Author response to Decision Letter 2]

24 Jun 2025

Reviewer #1: Comments to the Author:Title: Valorization of Various Lignocellulosic Wastes to Ganoderma lucidum (Reishi Mushroom) Cultivation and Their FT-IR AssessmentsOverview and general recommendation:Authors have made all needed improvements to their manuscript and are well thanked for that. Therefore, based on the overall evaluation of the manuscript, I find it well suitable for publication in current form.

We appreciate the positive feedback and are pleased that the reviewer finds our manuscript suitable for publication in its current form.

Reviewer #3: In this work by Caglar Akcay and colleagues, the cultivation of the medicinal mushroom Ganoderma lucidum was tested on various agricultural residues. The degradative capacity of G. lucidum on different substrates was assessed using Fourier-transform infrared (FTIR) spectroscopy. Furthermore, the chemical composition of both the substrates and the fruiting bodies obtained after cultivation was investigated.Overall, the manuscript requires substantial improvements, particularly in methodology, language and English style. Specifically:

Abstract

We thank the reviewer for their detailed and constructive comments. Below are our responses to each point:

Revisions are necessary to include results concerning the chemical composition of both the substrates and the harvested fruiting bodies.

Line 16: The full scientific name Ganoderma lucidum along with the authority should be provided at its first mention, both in the abstract and the main text. The common name "Reishi" should be inserted in parentheses immediately afterward and removed from line 18.Introduction –Line 40: The full name and authority for Ganoderma lucidum should be provided at first mention. Additionally, when the name appears at the beginning of a sentence (see line 56), it should be written out in full.

Updated to include the full scientific name with authority: Ganoderma lucidum (Curtis) P. Karst. The common name "Reishi" is now in parentheses at first mention.

Line 77: Replace “Turkiye” with “Turkey.”

Replaced with "Turkey" as requested

Lines 81–82: Insert the authority for Pleurotus ostreatus.

Added authority for Pleurotus ostreatus: (Jacq.) P. Kumm.

Lines 82–84: While the authors provided context for the selection of hazelnut branches as a substrate, no justification is given for the choice of rhododendron. Context should be added to support this choice.

Added context: Rhododendron sp. branches were selected due to their regional abundance as forestry waste, aligning with our goal to valorize underutilized lignocellulosic materials.

Materials and Methods –Line 97: Please insert the authority for Fagus orientalis.Lines 98–102: Move the substrate abbreviations to lines 96–98, where they are first mentioned.Line 102: Replace “Turkiye” with “Turkey.”

Lines 110–116: This section should be moved to the Introduction, as it explains the objectives of the study.

Actually, our aim here was to explain why we used these materials and their specific ratios in the study; it was a response to another reviewer. If possible, we would prefer to keep it here. Thank you for your suggestion

Line 127: Replace “Turkiye” with “Turkey.”Lines 127–128: Replace "compost" with "substrate."

Lines 138–139:

After colonization, were the bags opened to stimulate primordia formation, or were they kept closed until primordia emerged and then opened to promote primordia development and fruiting body expansion? Please clarify.

following sentence was added to the text

“All cultivation bags remained hermetically sealed during the entire process, with only the sterile cotton filter at the neck being removed to permit the directional growth and emergence of fruiting bodies”.

Lines 145–150: This information should be rewritten in a more discursive manner rather than presented as bullet point.

Rewritten as a discursive paragraph: Spawn run time was recorded as the duration for complete mycelial colonization, assessed via daily visual inspection. Earliness represented the period from inoculation to first harvest-ready fruiting body. Dry matter content was determined by oven-drying fresh mushrooms at 40°C to constant weight."

Line 154: What was the dry weight of the different formulated substrates? This is essential for calculating biological efficiency. Please provide this information.

Following sentence was added to the manuscript:

Substrates were initially weighed as dry materials (500 g per bag, except for high-volume CH and WS which were reduced to 250 g to accommodate bag capacity).

Lines 156–166: This paragraph should be written in a more discursive manner and should include more detailed information about the methods used.

Following text was added to the manuscript:

“The extractives content of both pre-cultivation control substrates (HB-C, HH-C, WS-C, RD-C, OW-C, BW-C, CH-C, CG-C) and post-cultivation fungal-degraded materials (HB-F, HH-F, WS-F, RD-F, OW-F, BW-F, CH-F, CG-F) was determined according to a modified TAPPI T 204 cm-17 standard. All substrate samples were first ground to 40-mesh particle size using a Wiley mill and then oven-dried at 103±2°C until constant weight was achieved. For the extraction process, precisely weighed 5 g aliquots of each dried sample were subjected to solvent extraction (toluene: acetone: ethanol mixture [4/1/1, (v/v)] in a Soxhlet apparatus for 6 hours, with the extraction cycles carefully monitored to ensure complete removal of soluble compounds. Following extraction, the solutions were vacuum-filtered through pre-weighed porosity-2 crucibles, and the retained residues were subsequently dried at 103±2°C for 12 hours. The extractives content was calculated gravimetrically by comparing the initial sample mass to the mass of insoluble residue after extraction, with all measurements performed in triplicate to ensure reproducibility[17].

The holocellulose content of the five substrate samples before and after cultivation (hazelnut pruning waste, hazelnut husk, rice husk, coffee grounds, and wheat straw) was determined using an optimized Wise and John chlorite method. Extractives-free 40-mesh samples (5 g), prepared by oven-drying at 103±2°C, were subjected to a four-stage delignification process in 250 mL Erlenmeyer flasks. Each reaction cycle consisted of adding 160 mL distilled water, 1.5 g sodium chlorite (NaClO₂), and 0.5 mL glacial acetic acid, followed by incubation at 78±1°C in a temperature-controlled water bath with continuous magnetic stirring. The mixture underwent vigorous shaking every 15 minutes during the 1-hour reaction period to ensure homogeneous treatment. After each cycle, fresh reagents (1.5 g NaClO₂ + 0.5 mL acetic acid) were added, with the complete process repeating four times to achieve thorough lignin removal. The resulting holocellulose was collected by vacuum filtration through pre-weighed porosity-2 glass crucibles, then sequentially washed with ice-cold distilled water (to remove residual acids) and acetone (to facilitate drying). The purified holocellulose was dried to constant weight at 103±2°C. The holocellulose content of the substrates (%) were determined relative to the initial full dry weight[18].

Alpha -cellulosecontent was determined following TAPPI T 203 cm-09 with modifications. Precisely 2 g of holocellulose was treated with 17.5% NaOH (10 mL initial + 2×5 mL additions at 5-min intervals) at 20.0±0.5°C for 30 min. After dilution with 33 mL distilled water and 60 min incubation, the mixture was filtered through porosity-2 crucibles. The residue was sequentially washed with 8.3% NaOH (100 mL), 10% acetic acid (15 mL), and distilled water (250 mL), then dried at 105°C to constant weight. Alpha-cellulose content was calculated gravimetrically relative to initial holocellulose mass, with triplicate measurements [19].

Ash content was calculated by combusting substrates at 575°C until constant weight (TAPPI T211 om-16) to determine inorganic residues. pH was measured in a 1:10 (w/v) substrate-water suspension using a calibrated pH meter”

Lines 192–196: Please provide additional details on how the obtained spectra were processed. How many replicates per treatment were conducted? The figures (Figures 1–9) suggest that baseline correction was not performed. To better highlight significant changes in the substrates before and after fungal growth, it is recommended that the authors include histograms of peak areas (%) processed using curve fitting in the region from 1800 to 1200 cm⁻¹.

Following text was added to the manuscript:

Line 249: FT-IR spectra were obtained using an IRPrestige-21 FT-IR Spectrophotometer (Shimadzu) equipped with a single-reflection ATR sampling module. The spectra were recorded in the range of 4000–400 cm-1 with a resolution of 4 cm-1 by averaging 32 scans. Spectral data in the region between 1800 and 800 cm-1 were processed using curve fitting analysis performed in the OriginPro 2013 software (OriginLab Corporation, Northampton, MA 01060 USA), by determining the area under each band. The FT-IR spectra were fitted using Gaussian-shaped bands. Optimal Gaussian curve fitting was determined when the best fit was achieved (reduced chi-square < 1×10-6) and based on the agreement between the calculated areas (R-square, R2 = 0.998–0.987 for substrates, R2 = 0.901-0.941 for mushrooms).

Using the Gaussian curve fitting procedure, peak areas for both the substrates and the fungi were analyzed, and the corresponding percentage histograms were incorporated into the revised Figures 1–9. As the histograms adequately represent the data, Table 5 has been omitted. Accordingly, the results presented between lines 433 and 631 have been revised in light of the updated histograms and spectral data.

Following text was added to the manuscript:

Line 433: The FT-IR spectra in the fingerprint region (1800–800 cm-1), which show the structural changes in hazelnut branches (HB), beech wood (BW), oak wood (OW), rhododendron branches (RD), corn husk (CH), wheat straw (WS), hazelnut husk (HH), and spent coffee grounds (CG) after fungal attack compared to the control samples (before fungal attack), along with the histograms of areas obtained through the Gaussian curve fitting procedure applied to the examined regions of these spectra, are given in Figs 1–8.

Line 446: Fig 1. FT-IR results. a) FT-IR spectra of HB-C and HB-F; b) histograms of FT-IR peak areas processed using Gaussian curve fitting.

Fig 2. FT-IR results. a) FT-IR spectra of BW-C and BW-F; b) histograms of FT-IR peak areas processed using Gaussian curve fitting.

Fig 3. FT-IR results. a) FT-IR spectra of OW-C and OW-F; b) histograms of FT-IR peak areas processed using Gaussian curve fitting.

Fig 4. FT-IR results. a) FT-IR spectra of RD-C and RD-F; b) histograms of FT-IR peak areas processed using Gaussian curve fitting.

Fig 5. FT-IR results. a) FT-IR spectra of WS-C and WS-F; b) histograms of FT-IR peak areas processed using Gaussian curve fitting.

Fig 6. FT-IR results. a) FT-IR spectra of CH-C and CH-F; b) histograms of FT-IR peak areas processed using Gaussian curve fitting.

Fig 7. FT-IR results. a) FT-IR spectra of HH-C and HH-F; b) histograms of FT-IR peak areas processed using Gaussian curve fitting.

Fig 8. FT-IR results. a) FT-IR spectra of CG-C and CG-F; b) histograms of FT-IR peak areas processed using Gaussian curve fitting.

Line 470-568:

The Gaussian curve fitting procedure applied to the FT-IR spectra of the substrates gave twelve Gaussian curves centered at 890, 1030, 1110, 1150, 1240, 1320, 1370, 1420, 1460, 1510, 1620, and 1730 cm-1. The percentage area of each band showed significant changes after fungal attack. The functional groups corresponding to these bands were examined in detail.

The bands observed between 1750 and 1720 cm-1 are attributed to hemicellulose degradation in lignocellulosic materials. This band indicates unconjugated C=O vibrations of acetyl, carboxylic acid, and uronic ester groups within the hemicellulose structure [31, 32]. Hemicellulose degradation caused by fungal attack resulted in a decrease in the area of this band (12) in all samples. The strong band observed at 1742 cm-1 in the CG-C sample is attributed to the C=O stretching vibrations of ester groups from kinic acid and lipids [33]. After the fungal attack, this band disappeared.

The bands between 1650 and 1600 cm-1 correspond to lignin degradation in lignocellulosic materials [31]. In this region, the deformation of O-H groups in the lignin structure (1640 cm-1), C=C and C=O stretching in the lignin aromatic chain (1630 cm-1), and C-O stretching vibrations in the lignin skeletal structure (1630 cm-1) overlapped, forming a broad band [16]. As a result of lignin degradation by G. lucidum, a significant increase in the area of this band (11) was observed in the HB, BW, CH, HH, and CG samples, whereas no notable change was detected in the OW, RD, and WS samples. This result may be associated with the selective lignin degradation by G. lucidum mycelia, depending on the type and structure of the substrate. The broad band observed at 1638 cm-1 in the CG-C sample is attributed to C=C stretching vibrations of lipids and fatty acids, C=O stretching vibrations of caffeine, and C=C, C=O, and C-O stretching vibrations of the lignin aromatic chain [34].

The bands between 1530 and 1500 cm-1 correspond to the aromatic C–O stretching vibrations of lignin [35]. Following fungal attack, the area of this band (10) increased in the CG sample, while it decreased in all other samples. Similarly, Akcay et al. [16] reported that Pleurotus ostreatus increased the visibility of the 1652 cm-1 band due to lignin degradation in coffee. However, in the present study, since fungal growth could not be achieved on the CG substrate, this may be associated with the lower impact of G. lucidum on lignin in spent coffee grounds compared to Pleurotus ostreatus.

The bands between 1465–1455 cm-1 and 1425–1410 cm-1 are attributed to C-H deformations of CH2 and CH3 groups in lignin and hemicellulose, as well as chlorogenic acids in coffee, and CH2 in-plane bending vibrations in cellulose and lignin, respectively [36–38]. After fungal attack, the area of the first band (9) decreased in the HB, BW, OW, RD, and CG samples, while it slightly increased in the WS, CH, and HH samples. The area of the second band (8), however, decreased in all samples. A similar result was observed for CG in Pleurotus ostreatus mycelia [16].

The bands between 1375–1360 cm-1 and 1330–1310 cm-1 correspond to symmetric and asymmetric C-H deformation vibrations of cellulose and hemicellulose, O-H deformation of kinic acid, CH₂ in-plane bending vibrations at the sixth carbon of crystalline cellulose, and C-H deformation of kinic acid, respectively [35,39]. After the fungal attack, the area of the first band (7) increased for the HH and CG samples, while it decreased for all other samples. Similarly, the area of the second band (6) increased for the CH, HH, and CG samples, but decreased in all others.

The bands between 1250–1230 cm-1 correspond to C-O stretching vibrations of hemicellulose, lignin, and kinic acid (in CG samples) [39]. After the fungal attack, the area of this band (5) decreased for the all samples.

The weak shoulders observed between 1165–1155 cm-1 and 1120–1075 cm-1, along with the band observed between 1040–1020 cm-1, are attributed to C–O–C vibrations of cellulose and hemicellulose (glycosidic linkages), quinic acid (in the CG sample), and C–OH bending vibrations [33,34,40]. After fungal attack, the area of the first band (4) decreased in the WS and CG samples, while it increased in the other samples. The significant decrease in the band area of the CG sample may be associated with the removal of quinic acids from the structure. The area of the second band (3) decreased in the HB, BW, RD, WS, and CH samples, while it increased in the OW, HH, and CG samples. The area of the third band (2) decreased in the HH sample, whereas it increased in all other samples.

The weak bands observed between 900 and 870 cm-1 are attributed to C1–O–C4 β-(1→4) glycosidic linkages [40,41]. The area of these bands (1) increased in the CH, HH, and CG samples, while it decreased in all other samples.

G. lucidum was observed to exhibit degradative effects on ligno

---

## [Decision Letter · Decision Letter 2]

Valorization of Various Lignocellulosic Wastes to Ganoderma lucidum (Curtis) P. Karst (Reishi Mushroom) Cultivation and Their FT-IR Assessments

PONE-D-25-12794R2

Dear Dr. ARSLAN,

We’re pleased to inform you that your manuscript has been judged scientifically suitable for publication and will be formally accepted for publication once it meets all outstanding technical requirements.

Kind regards,

Lee Seong

Academic Editor

PLOS ONE

Additional Editor Comments (optional):

Reviewers' comments:

Reviewer's Responses to Questions

**Comments to the Author**

1. If the authors have adequately addressed your comments raised in a previous round of review and you feel that this manuscript is now acceptable for publication, you may indicate that here to bypass the “Comments to the Author” section, enter your conflict of interest statement in the “Confidential to Editor” section, and submit your "Accept" recommendation.

Reviewer #1: All comments have been addressed

Reviewer #4: All comments have been addressed

2. Is the manuscript technically sound, and do the data support the conclusions?

Reviewer #1: Yes

Reviewer #4: Yes

3. Has the statistical analysis been performed appropriately and rigorously? 

Reviewer #1: Yes

Reviewer #4: (No Response)

4. Have the authors made all data underlying the findings in their manuscript fully available?

Reviewer #1: Yes

Reviewer #4: (No Response)

5. Is the manuscript presented in an intelligible fashion and written in standard English?

Reviewer #1: Yes

Reviewer #4: Yes

6. Review Comments to the Author

Reviewer #1: Comments to the Author:

Title: Valorization of Various Lignocellulosic Wastes to Ganoderma lucidum (Curtis) P. Karst (Reishi Mushroom) Cultivation and Their FT-IR Assessments

Overview and general recommendation:

Authors have made all needed improvements to their manuscript and are well thanked for that. Therefore, based on the overall evaluation of the manuscript, I find it well suitable for publication in current form.

Reviewer #4: Authors have satisfactorily addressed all the comments made on the previous version of this submission. I have no further comments on this.

7. PLOS authors have the option to publish the peer review history of their article (what does this mean? ). If published, this will include your full peer review and any attached files.

**Do you want your identity to be public for this peer review?** For information about this choice, including consent withdrawal, please see our Privacy Policy .

Reviewer #1: No

Reviewer #4: No

---

## [Editor Report · Acceptance letter]

PONE-D-25-12794R2

PLOS ONE

Dear Dr. Arslan,

I'm pleased to inform you that your manuscript has been deemed suitable for publication in PLOS ONE. Congratulations! Your manuscript is now being handed over to our production team.

Kind regards,

on behalf of

Dr. Lee Seong

Academic Editor

PLOS ONE